# Depth as Modulation in Weight-Sharing Transformers

## Abstract

Weight sharing reduces Transformer parameters by reusing a single block across depth, but it enforces a depth-invariant transformation that can limit specialization. We study a simple way to reintroduce depth variation in weight-sharing (recurrent) Transformers: we freeze the shared block and train small, depth-indexed low-rank modules inside multi-head self-attention (MHSA). We consider two parameterizations: QKOV-LoRA applies depth-indexed Low-Rank Adaptation (LoRA) to the $Q, K, V, O$ projections, while QK/OV-Circuit applies low-rank corrections to the QK routing and OV aggregation operators, yielding a more constrained update with different compute trade-offs. Under matched trainable-parameter budgets, depth-indexed MHSA modulation tends to recover accuracy in the vision settings we studied, with particularly strong gains in low-data regimes; in language, the effects are more task-dependent and include both improvements and degradations. Overall, the results clarify when depth-wise MHSA modulation is a useful complement to weight sharing and how it trades off accuracy and efficiency.

## 1 Introduction

Transformer architectures have become central in language and vision, and their representations evolve with depth (Vaswani et al., 2017; Dosovitskiy et al., 2021). Empirical analyses suggest that early layers capture more local interactions or low-level features, while later layers integrate broader context and shape representations for downstream tasks (Raghu et al., 2021; Geshkovski et al., 2023). These findings motivate treating depth as an axis of specialization rather than simple repetition.

A direct way to reduce the parameter count of deep Transformers is to share parameters across layers. Models such as the Universal Transformer (Dehghani et al., 2019) and ALBERT (Lan et al., 2020) demonstrate that repeatedly applying a shared block can yield compact models with competitive performance. However, this strategy introduces a trade-off. While parameter sharing achieves efficiency, it imposes a uniform transformation at all depths, which conflicts with the observed need for functional specialization. A single, fixed operator can be too restrictive for the multi-stage process of representation refinement that unshared models learn. Although some approaches add small, fixed modules to a shared backbone (Zhang et al., 2022; Rajabzadeh et al., 2024), the modification remains constant across depth and may not address this limitation.

**Positioning vs. prior work.** Parameter sharing across depth has been explored in the Universal Transformer and ALBERT; in vision, compression methods further leverage weight reuse in conjunction with per-layer transformations or prompts (Zhang et al., 2022; Jia et al., 2022). We study depth-indexed MHSA modulation in weight-sharing Transformers and compare it to depth-invariant adapters. We target either the projection matrices (QKOV-LoRA) or the composite operators that govern attention computation (QK/OV-Circuit). We do not claim general superiority over all adapter families; rather, we evaluate when depth-dependent modulation is beneficial and how it trades off accuracy and parameter cost.

To address the uniformity constraint, we propose a layer-wise modulation architecture for weight-sharing Transformers. We keep a shared backbone and add small, depth-indexed trainable modules that modulate MHSA at each depth. This retains the efficiency benefits of weight sharing while allowing the transformation to vary across layers.

We target modulation in the Multi-Head Self-Attention (MHSA) mechanism, which governs inter-token relationships. Analyses show that attention drives the geometric organization of the feature space, for example by grouping semantically

similar tokens into clusters (Geshkovski et al., 2025; Raghu et al., 2021). This consolidation is particularly important in deeper layers. We modulate the projections within MHSA ($Q, K, O, V$) to directly influence both the routing pattern of token interactions and the aggregation of information at each layer.

We implement this idea with depth-indexed Low-Rank Adaptation (LoRA) modules (Hu et al., 2022) inside MHSA in two variants: QKOV-LoRA, which modulates the $Q, K, O, V$ projections, and QK/OV-Circuit, a circuit-inspired parameterization (Elhage et al., 2021) that modulates the QK and OV composite operators.

We make three contributions. (1) We formalize the layer-wise modulation architecture described above. (2) We develop two concrete MHSA-based implementations, QKOV-LoRA and QK/OV-Circuit, with complementary efficiency profiles. (3) Through experiments on vision and language benchmarks, we evaluate when depth-dependent MHSA modulation improves over standard weight sharing and characterize accuracy and parameter trade-offs.

The remainder of the paper is organized as follows. We review related work in Section 2, provide background and conceptual motivation in Section 3, detail our architectures in Section 4, present experiments in Section 5, and conclude in Section 6.

## 2 Related Work

We review work on parameter sharing, depth-wise specialization, and parameter-efficient adaptation. This context frames our goal: keep the efficiency of a shared backbone while allowing depth-dependent behavior through small trainable modules.

**Parameter Sharing and the Uniformity Constraint.**   To mitigate the memory and computational costs of deep Transformers, researchers have explored parameter sharing across layers. Models such as the Universal Transformer (Dehghani et al., 2019) and ALBERT (Lan et al., 2020) demonstrated that repeatedly applying a shared block can yield compact models with competitive performance. Subsequent work explored more granular variants, including sharing only middle layers (Reid et al., 2021) or sharing at the level of attention heads (Cao et al., 2024). However, a trade-off emerges: efficiency is gained at the cost of imposing a uniform transformation at all depths. This uniformity conflicts with the behavior of standard Transformers, where functional specialization evolves through the layers. Our work retains the shared backbone for efficiency and studies depth-indexed MHSA modulation as an approach to mitigate the uniformity constraint. In vision, MiniViT compresses Vision Transformers (ViTs) via weight multiplexing, providing another approach to parameter efficiency (Zhang et al., 2022). Relaxed Recursive Transformers are closely related to our setting: they add layer-wise LoRA to recurrent/shared-block Transformers and retrain the model end-to-end (Bae et al., 2025). In contrast, our approach (Section 4) freezes the shared block and trains only depth-indexed MHSA modules, focusing modulation on token interactions.

**The Case for Depth-wise Functional Specialization.**   Prior work suggests that a Transformer's depth supports a sequence of functionally distinct processing stages. In language models, lower layers capture surface and syntactic patterns, while upper layers compose semantic meaning (Tenney et al., 2019; Jawahar et al., 2019; Rogers et al., 2020; Hewitt & Manning, 2019; Ethayarajh, 2019; Clark et al., 2019; Geva et al., 2021). In vision, ViTs exhibit a shift from diverse low-level features to integrated, consolidated representations in later layers; attention maps increasingly highlight object-level structure (Raghu et al., 2021; Caron et al., 2021). This consolidation supports token merging or pruning with little accuracy loss (Bolya et al., 2023; Rao et al., 2021). Together, these findings support the view that depth benefits from functionally distinct transformations rather than a single fixed operator. Such layer-wise changes are often quantified with representation similarity measures such as (linear) centered kernel alignment (CKA) (Kornblith et al., 2019).

**Lightweight Adapters as a Mechanism for Specialization.**   Parameter-efficient transfer learning introduced lightweight adapters (Houlsby et al., 2019), and low-rank updates such as LoRA provide an alternative with similar goals (Hu et al., 2022). These methods are commonly compared on suites such as GLUE (Wang et al., 2018). Recent work also considers sharing low-rank modules within attention (e.g., across Q/K/V paths within a single ViT layer) (Zhong et al., 2025), which is a different notion of sharing than depth-dependent modulation across repeated layers. Our approach differs by targeting depth-indexed modulation within MHSA of a frozen shared block; Section 4 gives details.

## 3 Background

We summarize three perspectives that provide context for our approach: deep networks as discretized dynamical systems, empirical evidence for depth-wise functional specialization in Transformers, and design approaches for parameter-efficient adaptation. Together, they inform a shared backbone augmented with small, depth-indexed modulation modules.

### 3.1 Transformers as Discretized Dynamical Systems

Deep residual networks can be viewed as discretizations of continuous-time ordinary differential equations (ODEs) (E, 2017; Chen et al., 2018). In this view, the forward pass corresponds to numerically integrating a learned vector field over time (depth), with each layer representing a time step. This view supports analyses of stability, convergence, and expressivity using tools from numerical analysis and dynamical systems (Haber & Ruthotto, 2018).

For Transformers, one related formulation models the token set as an interacting particle system (Geshkovski et al., 2023). Let $x_i(t) \in \mathbb{R}^d$ denote the representation of token $i$ at continuous depth $t$. A generic ODE decomposition separates inter-token interactions (MHSA) and per-token nonlinearity (FFN):

$$\frac{\mathrm{d}x_i(t)}{\mathrm{d}t} = \underbrace{F\big(x_i(t), \{x_j(t)\}_{j=1}^N, t\big)}_{\text{interaction via MHSA (diffusion)}} + \underbrace{G\big(x_i(t), t\big)}_{\text{self-dynamics via FFN (convection)}} . \tag{1}$$

A standard Transformer block can be interpreted as an operator-splitting scheme such as Lie–Trotter that alternates these two operators (Lu et al., 2019).

The interaction term $F$ can be related to the *Transformer Circuits* decomposition (Elhage et al., 2021), which separates attention into a routing sub-circuit (QK) and a content sub-circuit (OV). Writing a single-head form for clarity,

$$F_i(t) = \sum_{h=1}^H \frac{1}{L_{i,h}} \sum_{j=1}^N \exp\left( \underbrace{\frac{x_i^\top(t)\, W_Q^{h\top}(t)\, W_K^h(t)\, x_j(t)}{\sqrt{d_k}}}_{\text{QK circuit: information routing}} \right) \underbrace{\big(W_O^h(t)\, W_V^h(t)\, x_j(t)\big)}_{\text{OV circuit: content aggregation}}, \tag{2}$$

where $L_{i,h}$ is the softmax normalizer. Modulating the QK circuit changes the data-dependent interaction topology (routing), while modulating the OV circuit changes how information is transformed and aggregated. This separation provides two levers for controlling attention computation.

### 3.2 Empirical Evidence for Depth-wise Functional Specialization

Prior work reports that layers in deep Transformers are not redundant; they form a pipeline of related yet distinct computations.

**Language.** Probing studies on BERT-like models show that lower layers encode surface and syntactic information, while higher layers compose more abstract semantics (Tenney et al., 2019; Jawahar et al., 2019; Rogers et al., 2020). The "structural probe" reveals that syntactic structure is linearly recoverable from middle layers (Hewitt & Manning, 2019). Contextualization increases with depth, with upper layers exhibiting lower self-similarity and higher anisotropy (Ethayarajh, 2019). Attention heads specialize in roles such as delimiter detection or dependency tracking (Clark et al., 2019), and FFNs store abstract patterns (Geva et al., 2021).

**Vision.** In ViTs, early layers extract diverse low-level features, while later layers consolidate and integrate information (Raghu et al., 2021). In self-supervised ViTs such as DINO, later attention heads often highlight object-level structure (Caron et al., 2021). Consolidation implies redundancy among tokens, enabling merging or pruning in late layers with minimal accuracy loss (Bolya et al., 2023; Rao et al., 2021).

These findings support treating depth as a sequence of related but distinct transformations, rather than repeated applications of a single fixed operator.

### 3.3 Late-Layer Dynamics: Refinement, Redundancy, and Residual Connections

Residual connections help propagate features forward across depth (He et al., 2016; Vaswani et al., 2017). Update magnitudes typically shrink with depth, and adjacent late layers often show high representation similarity as measured by CKA (Kornblith et al., 2019). This has been interpreted as redundancy, where many heads or layers can be pruned with limited impact (Voita et al., 2019; Michel et al., 2019; Dalvi et al., 2020). Late layers can also perform fine-grained refinement. Adaptive computation mechanisms such as ACT in Universal Transformers allocate more steps to harder inputs (Dehghani et al., 2019; Xin et al., 2020). Vision architectures like CaiT specialize late blocks with class attention to guide consolidation (Touvron et al., 2021b).

### 3.4 Parameter Sharing Across Depth and Layer-wise Modulation

Parameter sharing across depth is an effective route to memory- and compute-efficient Transformers, as in the Universal Transformer and ALBERT (Dehghani et al., 2019; Lan et al., 2020). In a recurrent/shared-block setting, the same block is applied at every depth step, which makes the update depth-invariant. This differs from the layer-by-layer changes commonly reported in standard Transformers (Tenney et al., 2019; Jawahar et al., 2019; Raghu et al., 2021).

We introduce layer-wise low-rank modulation: small, learnable modules indexed by depth. In this paper, "layer-wise" means these modules are not shared across depth steps, even though the backbone is shared. We focus modulation on MHSA so that both routing (QK) and content aggregation (OV) can vary across depth with a compact parameter budget. We instantiate this with QKOV-LoRA (LoRA on $Q, K, O, V$) and QK/OV-Circuit (low-rank modulation on QK and OV composite operators).

## 4 Method: Layer-wise Modulation for Weight-Sharing Transformers

Weight sharing reduces the parameter count of deep Transformers by reusing a shared block across depth, but it applies the same update rule at every depth. We relax this constraint by adding small, depth-indexed modulation modules inside MHSA while keeping the shared backbone frozen. We define the framework and two instantiations, QKOV-LoRA and QK/OV-Circuit.

### 4.1 Framework

We simulate a deep $L$-layer Transformer by iteratively applying a single pretrained encoder block $\mathcal{B}_{\text{shared}}$. We allow a small, depth-indexed module $\delta_i$ to modify the shared block at effective depth $i$:

$$h_{i+1} = \mathcal{B}_{\text{shared}}(h_i; \theta_{\text{frozen}}, \delta_i), \quad \text{for } i \in \{l, \ldots, L\}, \tag{3}$$

where $h_i$ denotes the representation at effective depth $i$. We keep $\theta_{\text{frozen}}$ fixed and train only the small modules $\delta_i$, which are unique to each effective depth. We apply $\delta_i$ only within MHSA and keep FFN frozen so that the modification is confined to the attention computation. We consider two parameterizations: (i) low-rank updates on the projection weights (QKOV-LoRA) and (ii) low-rank corrections applied at the composite sites used to form attention logits and the OV contraction (QK/OV-Circuit).

We modulate MHSA because it governs how tokens interact. We keep FFN fixed to avoid conflating attention changes with concurrent changes to per-token nonlinear transformations.

### 4.2 Architectures

We instantiate the same framework for language and vision backbones.

#### 4.2.1 Language: ALBERT

For NLP, we adapt the ALBERT architecture, which uses a recurrent structure with a single shared encoder block. This offers strong parameter savings but applies a depth-invariant transformation across layers. Starting with a pretrained ALBERT-base-v2, we maintain its recurrent application of the shared block. For the initial $l$ steps, the model is unchanged.

From effective depth $l + 1$ onwards, we introduce depth-indexed MHSA modules $\delta_i$. This choice is motivated by evidence that deeper layers support more global integration and refinement (Section 3.2). We evaluate depth-indexed modulation against depth-invariant baselines in Section 5.

### 4.2.2 Vision: ViT

In contrast to ALBERT, standard Vision Transformers (ViTs) are deep, non-recurrent stacks of unique layers. Directly applying a single shared block from layer 1 can be suboptimal, as early and late layers can serve different roles (feature extraction vs. integration). Our design is guided by the empirical observation of late-layer representational similarity (Section 3.3). CKA analyses suggest that while early layers are highly specialized, later layers become more functionally redundant as they iteratively refine a global representation.

Based on this insight, we propose the hybrid architecture in Figure 2. We take a pretrained ViT-B/16 and freeze its first $l$ layers. The $l + 1$-th encoder block becomes the recurrent shared block $\mathcal{B}_{\text{shared}}$ and is applied iteratively to replace the original layers from $l + 1$ to $L$. At each recurrent step, its MHSA module is modulated by a depth-indexed module $\delta_i$.

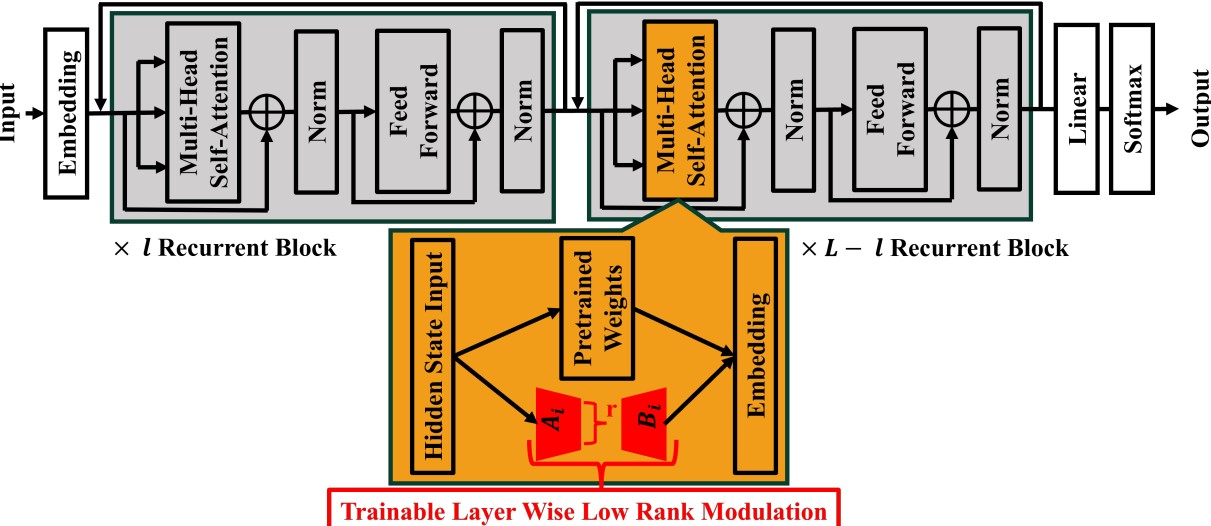

Figure 1: **Layer-wise MHSA Modulation in a Recurrent NLP Model.** We enhance ALBERT by adding a depth-indexed LoRA module $(A_i, B_i)$ inside MHSA at each recurrent step $i \geq l + 1$, while keeping the shared block frozen.

### 4.3 Depth-indexed LoRA Modules

We implement $\delta_i$ using Low-Rank Adaptation (LoRA) (Hu et al., 2022). LoRA modifies a pretrained weight matrix $W_0$ with a low-rank update $W = W_0 + BA$, where only the small matrices $A$ and $B$ are trainable. We investigate two MHSA-focused strategies. For a weight $W \in \mathbb{R}^{d_{\text{out}} \times d_{\text{in}}}$, the update uses $A \in \mathbb{R}^{r \times d_{\text{in}}}$ and $B \in \mathbb{R}^{d_{\text{out}} \times r}$. Unless stated otherwise, the rank $r$ is shared across heads and projections.

### 4.3.1 QKOV-LoRA

This first strategy, QKOV-LoRA, aims for maximum flexibility by applying independent LoRA modules to all four projection matrices $(W_Q, W_K, W_O, W_V)$ of the MHSA. For each effective layer $i \geq l + 1$, this yields four distinct updates:

$$W_{Q,i} = W_{Q,\text{shared}} + B_{Q,i}A_{Q,i}, \qquad\qquad W_{K,i} = W_{K,\text{shared}} + B_{K,i}A_{K,i} \qquad (4)$$

$$W_{O,i} = W_{O,\text{shared}} + B_{O,i}A_{O,i}, \qquad\qquad W_{V,i} = W_{V,\text{shared}} + B_{V,i}A_{V,i} \qquad (5)$$

This applies LoRA modules to the $Q, K, O, V$ projections, which allows both routing (via $Q$ and $K$) and content aggregation (via $O$ and $V$) to vary across effective depth. The attention logits use $Q_i K_i^\top / \sqrt{d_k}$ with $Q_i = X W_{Q,i}$ and

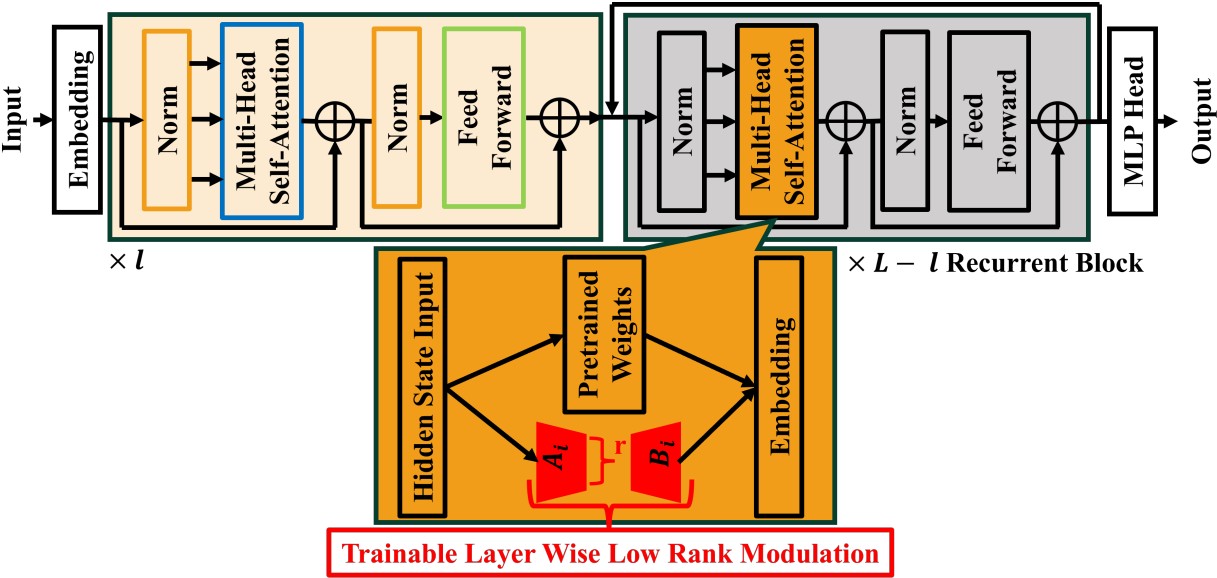

Figure 2: **Layer-wise MHSA Modulation in a ViT with a Recurrent Shared Block.** We freeze the first $l$ layers of a pretrained ViT. The $(l+1)$-th block is reused as a recurrent shared block, and we add a depth-indexed LoRA module $(A_i, B_i)$ inside MHSA at each step.

$K_i = XW_{K,i}$; the value path uses $O_iV_i$ with $O_i = XW_{O,i}$ and $V_i = XW_{V,i}$. Each projection adds a low-rank update with rank $r$.

### 4.3.2 QK/OV-Circuit

Our second strategy is inspired by the Transformer Circuits decomposition (Section 3.1). Instead of separate updates to $Q, K, O, V$, it applies low-rank corrections at two composite sites: the query–key (QK) term that determines routing, and the output–value (OV) term that determines the content path. At every layer $i \geq l+1$, we apply:

$$(W_{Q,i})^\top W_{K,i} \approx (W_{Q,\text{shared}})^\top W_{K,\text{shared}} + B_{QK,i}A_{QK,i}. \tag{6}$$

$$W_{O,i}W_{V,i} \approx W_{O,\text{shared}}W_{V,\text{shared}} + B_{OV,i}A_{OV,i}. \tag{7}$$

We insert low-rank corrections directly at (i) the logit site, $S_i \leftarrow S_i + \Delta S_i$ with $\Delta S_i = \hat{B}_{QK,i}\hat{A}_{QK,i}$, and (ii) the OV contraction, $(OV)_i \leftarrow (OV)_i + \Delta(OV)_i$ with $\Delta(OV)_i = \hat{B}_{OV,i}\hat{A}_{OV,i}$. This does not imply a unique pair $(W_{Q,i}, W_{K,i})$ that realizes the same composite; we therefore treat $\Delta S_i$ and $\Delta(OV)_i$ as independent, trainable low-rank modules with gradients applied at their insertion sites. For $n$ tokens per head, the learned correction factors include the token dimension in their shape: $\hat{A}_{QK,i} \in \mathbb{R}^{r \times n}$ and $\hat{B}_{QK,i} \in \mathbb{R}^{n \times r}$ (and analogously for OV after projection). Because these are stored parameters whose dimensions depend on $n$, the trained correction is tied to the sequence length used during training, which is a consideration when using the method with variable-length inputs. Parameters scale with $r$ and head count but avoid four separate projection updates.

## 5 Experiments

We evaluate layer-wise low-rank modulation for weight-sharing Transformers. We compare QKOV-LoRA and QK/OV-Circuit to depth-invariant baselines, and we match the number of trainable parameters wherever this is feasible.

We report three components: (i) language accuracy on GLUE and SuperGLUE, (ii) a layer-wise representation analysis in the language experiments to discuss tasks where depth-dependent modulation helps and tasks where it hurts, and (iii) vision accuracy on ImageNet-1k (including a low-data regime) and downstream transfer on CIFAR-10.

## 5.1 Experimental Setup

**Benchmarks and Models.** Our evaluation spans language and vision. For natural language understanding (NLU), we evaluate ALBERT-base-v2 (Lan et al., 2020) on GLUE (Wang et al., 2018) and SuperGLUE (Wang et al., 2019) and report accuracy. For vision, we evaluate a pretrained ViT-B/16 (Dosovitskiy et al., 2021) on ImageNet-1k (Deng et al., 2009) and report downstream transfer on CIFAR-10 (Krizhevsky et al., 2009).

**Models Under Comparison.** Our primary models are **QKOV-LoRA** and **QK/OV-Circuit**. In NLU, we compare against **LoRA** (Hu et al., 2022) and **Adapter** (Houlsby et al., 2019). In vision, we include baselines that reuse a shared ViT block across layers (simple weight sharing), as well as a depth-invariant LoRA baseline where the same LoRA module is applied to the repeated portion of the model at every recurrence step. We also include MiniViT (Zhang et al., 2022) as a related efficiency-oriented baseline.

**Implementation Details.** To ensure fair and reproducible comparisons, all experiments use AdamW (Loshchilov & Hutter, 2019) with a cosine learning-rate schedule. For GLUE and SuperGLUE, we report accuracy, mean and standard deviation over 10 random seeds, and Holm-corrected significance marks when tested (Holm, 1979). For vision experiments, we report accuracy, mean and standard deviation over 5 random seeds. We report trainable parameters for all methods. For LoRA-based methods, we additionally report the LoRA rank and the layer index at which modulation starts (from layer $l + 1$), as defined in Section 4 and illustrated in Figures 1 and 2. Full hyperparameters, learning rates, and training configurations are provided in the Appendix. We plan to release the source code upon publication.

## 5.2 Results on Natural Language Understanding

Table 1: Model parameters and computational costs.

| Method | Params (k) | Rank | Start Index | GFLOPs | Inference Throughput (samples/s) | Training Time (min/epoch) |
|---|---|---|---|---|---|---|
| LoRA | 101 | 16 | - | 44.22 | 621.0 | 1.12 |
| Adapter | 101 | - | - | 44.21 | 640.3 | 1.06 |
| **QKOV-LoRA** | 99.8 | 4 | 9 | 43.66 | 643.3 | 0.32 |
| **QK/OV-Circuit** | 101 | 6 | 3 | 61.10 | 365.2 | 2.45 |

Table 2: GLUE task performance. Cells marked with [†] indicate Holm-corrected $p < 0.05$ versus LoRA. This table reports the standard 8-task subset. Rank, trainable parameters, and Start Index settings are consolidated in Table 1.

| Method | CoLA | SST-2 | MRPC | QQP | STS-B | MNLI | QNLI | RTE | Avg. |
|---|---|---|---|---|---|---|---|---|---|
| LoRA | $52.7 \pm 2.0$ | $91.8 \pm 0.40$ | $\mathbf{90.7 \pm 0.95}$ | $85.1 \pm 0.20$ | $\mathbf{90.0 \pm 0.40}$ | $\mathbf{84.1 \pm 0.15}$ | $\mathbf{91.6 \pm 0.24}$ | $72.7 \pm 2.3$ | 82.3 |
| Adapter | $51.4 \pm 1.4$ | $91.9 \pm 0.42$ | $90.9 \pm 0.73$ | $84.4 \pm 0.07^\dagger$ | $90.3 \pm 0.27$ | $84.0 \pm 0.20$ | $91.3 \pm 0.23$ | $72.4 \pm 1.9$ | 82.1 |
| **QKOV-LoRA** | $\mathbf{79.2 \pm 0.74}^\dagger$ | $91.6 \pm 0.23$ | $86.4 \pm 1.2^\dagger$ | $\mathbf{87.6 \pm 0.15}^\dagger$ | $89.3 \pm 0.31^\dagger$ | $82.7 \pm 0.16^\dagger$ | $90.7 \pm 0.19^\dagger$ | $72.9 \pm 1.2$ | **85.1** |
| **QK/OV-Circuit** | $78.1 \pm 1.0^\dagger$ | $91.6 \pm 0.43$ | $83.8 \pm 1.9^\dagger$ | $\mathbf{87.7 \pm 0.11}^\dagger$ | $89.0 \pm 0.45^\dagger$ | $82.7 \pm 0.16^\dagger$ | $90.4 \pm 0.17^\dagger$ | $68.4 \pm 3.2^\dagger$ | 84.0 |

Table 3: SuperGLUE task performance. Cells marked with [†] indicate Holm-corrected $p < 0.05$ versus LoRA. This table reports the 7-task subset. Rank, trainable parameters, and Start Index settings are consolidated in Table 1.

| Method | BoolQ | CB | COPA | MultiRC | RTE | WiC | WSC | Avg. |
|---|---|---|---|---|---|---|---|---|
| LoRA | $70.1 \pm 1.0$ | $66.7 \pm 0.80$ | $61.7 \pm 4.6$ | $\mathbf{77.4 \pm 0.40}$ | $70.6 \pm 2.5$ | $\mathbf{70.5 \pm 1.2}$ | $58.0 \pm 3.3$ | 67.9 |
| Adapter | $\mathbf{75.6 \pm 0.63}^\dagger$ | $\mathbf{78.6 \pm 4.5}^\dagger$ | $61.6 \pm 3.9$ | $78.3 \pm 0.52$ | $73.1 \pm 1.8$ | $68.5 \pm 1.2$ | $54.2 \pm 5.2$ | **70.0** |
| **QKOV-LoRA** | $72.6 \pm 0.10^\dagger$ | $\mathbf{81.0 \pm 2.2}^\dagger$ | $63.0 \pm 1.6$ | $74.3 \pm 0.50^\dagger$ | $72.0 \pm 1.5$ | $66.1 \pm 1.1^\dagger$ | $53.8 \pm 2.8$ | 69.0 |
| **QK/OV-Circuit** | $69.9 \pm 0.58$ | $71.1 \pm 3.8$ | $59.9 \pm 4.2$ | $75.7 \pm 0.72^\dagger$ | $68.4 \pm 2.9^\dagger$ | $67.2 \pm 1.3^\dagger$ | $55.3 \pm 4.6$ | 66.8 |

We evaluate ALBERT-base-v2 on GLUE and SuperGLUE with four parameter-efficient methods: LoRA, Adapter, QKOV-LoRA, and QK/OV-Circuit. Adapter inserts bottleneck modules after the attention and FFN sublayers (Houlsby et al., 2019). To reduce confounding from adaptation budget, we align the number of trainable parameters as closely as

possible across methods. Table 1 summarizes trainable parameters, ranks (for LoRA-based methods), and the Start Index. Start Index denotes the first layer where depth-indexed modulation is activated (layer $l + 1$ in Section 4). In this configuration, LoRA uses rank 16, QKOV-LoRA uses rank 4 with Start Index 9, and QK/OV-Circuit uses rank 6 with Start Index 3. We use one Start Index per method in these tables, although the optimal value can vary by task. In these NLU tables, our primary comparisons match trainable parameters rather than LoRA rank, since different adaptation mechanisms use different parameterizations and rank alone is not a uniform measure of adaptation budget. We report rank explicitly for LoRA-based methods. For a rank-matched comparison (all rank 4), see Appendix Table 6, where depth-indexed methods also show gains over the LoRA baseline. A full fixed-rank sweep for NLU across all methods remains future work, and can be added if needed.

Table 1 also reports GFLOPs, inference throughput, and wall-clock training time. Throughput (samples/s) and training time (min/epoch) are measured on BoolQ with a maximum sequence length of 256 (batch size 16). Under this setting, QKOV-LoRA trains faster per epoch (0.32 min/epoch) than LoRA (1.12) and Adapter (1.06) while keeping similar GFLOPs and throughput. QK/OV-Circuit is slower and more compute-heavy (61.10 GFLOPs; 365.2 samples/s; 2.45 min/epoch). One possible factor is that its correction includes terms whose cost can depend on sequence length (Section 4), so we treat this as a compute trade-off in these experiments.

Tables 2 and 3 report mean $\pm$ std over 10 seeds. We mark a cell with $\dagger$ when the Holm-corrected $p$ value versus LoRA is below 0.05, and the mark can indicate either a higher or a lower value relative to LoRA. On GLUE, QKOV-LoRA is higher than LoRA on CoLA ($79.2\pm0.74^{\dagger}$ vs $52.7\pm2.0$, +26.5 points) but lower on MRPC ($86.4\pm1.2^{\dagger}$ vs $90.7\pm0.95$, -4.3 points). QK/OV-Circuit shows a similar pattern on these two tasks (CoLA: $78.1\pm1.0^{\dagger}$; MRPC: $83.8\pm1.9^{\dagger}$). On SuperGLUE, QKOV-LoRA is higher than LoRA on CB ($81.0\pm2.2^{\dagger}$ vs $66.7\pm0.80$, +14.3 points) but lower on WiC ($66.1\pm1.1^{\dagger}$ vs $70.5\pm1.2$, -4.4 points). QK/OV-Circuit is also lower on WiC ($67.2\pm1.3^{\dagger}$ vs $70.5\pm1.2$). In terms of task averages, QKOV-LoRA has the highest GLUE average (85.1), while Adapter has the highest SuperGLUE average (70.0).

These gains and drops are concentrated on a subset of tasks. In particular, QKOV-LoRA is higher than LoRA on CoLA (+26.5 points) and CB (+14.3 points), but lower on MRPC (-4.3 points) and WiC (-4.4 points) (Tables 2 and 3). To analyze this contrast, we compare a gain case (CoLA) and a drop case (WiC) using two descriptive measures: (i) layer-wise Linear CKA and (ii) token cosine similarity distributions within each layer (Section 5.3).

## 5.3 Representational Analysis: Linear CKA and Token Similarity

We report two complementary analyses on CoLA and WiC. **Protocol.** For Linear CKA (Kornblith et al., 2019), we sample up to 500 validation examples per task and compute a $12 \times 12$ pairwise similarity matrix across layers. For token cosine similarity, we sample up to 100 validation examples per task, subsample at most 1,000 token vectors per layer, normalize each token vector to unit L2 norm, and build 50-bin histograms over $[-1, 1]$ from all token-pair cosine similarities. Additional CKA results for tasks with notable performance differences are provided in Appendix A.5.

The first measure is Linear Centered Kernel Alignment (Linear CKA) (Kornblith et al., 2019), which quantifies the similarity between two layers' representation matrices. Given matrices $X \in \mathbb{R}^{n \times d}$ and $Y \in \mathbb{R}^{n \times d}$ from two layers (each containing $n$ examples and $d$ features), Linear CKA is defined as $\text{CKA}(X, Y) = \|Y^{\top}X\|_F^2 / (\|X^{\top}X\|_F \cdot \|Y^{\top}Y\|_F)$. This index takes values in $[0, 1]$, with larger values indicating more similar representational structure. We compute the $12 \times 12$ pairwise CKA matrix for each method and task. Phang et al. (2021) applied the same analysis to fine-tuned ALBERT and RoBERTa models and reported layer-wise clustering; here we use Linear CKA to compare depth-invariant and depth-dependent adaptation under weight sharing.

The second measure is the distribution of pairwise cosine similarities among all token vectors within each layer. Ethayarajh (2019) showed that contextualized representations in BERT can become increasingly anisotropic (concentrated in a narrow cone) in upper layers, and higher average pairwise cosine similarity reflects stronger directional concentration. We adopt a similar protocol: for a given layer, we compute the cosine similarity between all token-vector pairs and visualize the resulting distribution across layers. For this analysis, we compare LoRA (depth-invariant baseline) and QKOV-LoRA (depth-varying proposed method), as QKOV-LoRA exhibited the largest performance difference relative to LoRA across the two target tasks.

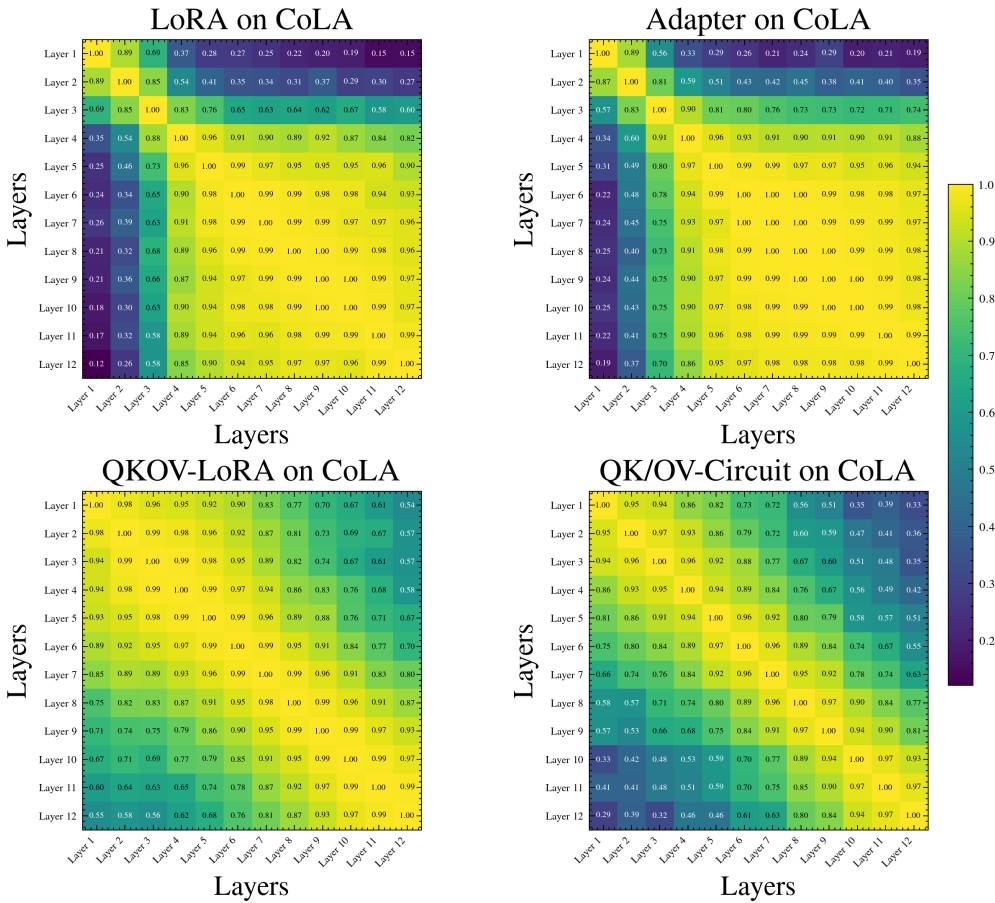

Figure 3: Layer-wise Linear CKA similarity on **CoLA** (gain case). Depth-invariant LoRA (rank 16) and Adapter show a late-layer plateau, with CKA between layers 5–12 concentrated between 0.91 and 1.00. Depth-dependent QKOV-LoRA (rank 4, Start Index 9) and QK/OV-Circuit (rank 6, Start Index 3) show a continuous decay of CKA with layer distance across the full depth range. In particular, QK/OV-Circuit reaches CKA 0.29 between Layer 1 and Layer 12, indicating the strongest depth-wise differentiation among the compared methods.

### 5.3.1 Linear CKA

Figures 3 and 4 show the $12 \times 12$ Linear CKA matrices for the two depth-invariant baselines (LoRA and Adapter) and the two depth-dependent methods (QKOV-LoRA and QK/OV-Circuit) on CoLA and WiC.

**CoLA.** On CoLA, depth-invariant LoRA and Adapter show a consistent two-phase pattern: earlier layers are more differentiated, while later layers become nearly identical (CKA between layers 5–12 concentrated between 0.91 and 1.00). This is consistent with weight sharing imposing a uniform update rule on the repeated portion of the model. In contrast, QKOV-LoRA and QK/OV-Circuit show a continuous decay of CKA with layer distance across depth, indicating that each layer executes a representation update that differs from the preceding layers. QK/OV-Circuit yields the strongest depth-wise differentiation (Layer 1 vs. Layer 12: CKA 0.29).

**WiC.** On WiC, QKOV-LoRA shows much weaker depth-wise differentiation than on CoLA (Layer 1 vs. Layer 12: CKA 0.85 compared to 0.55). QK/OV-Circuit shows the same trend (WiC: 0.72 vs. CoLA: 0.29). Together with the WiC drop in Table 3, these results are consistent with depth-dependent modulation failing to induce effective depth-wise differentiation on this task after training. In contrast, LoRA and Adapter show a similar two-phase pattern across tasks, consistent with depth-invariant adapters having limited ability to alter late-layer representation trajectories.

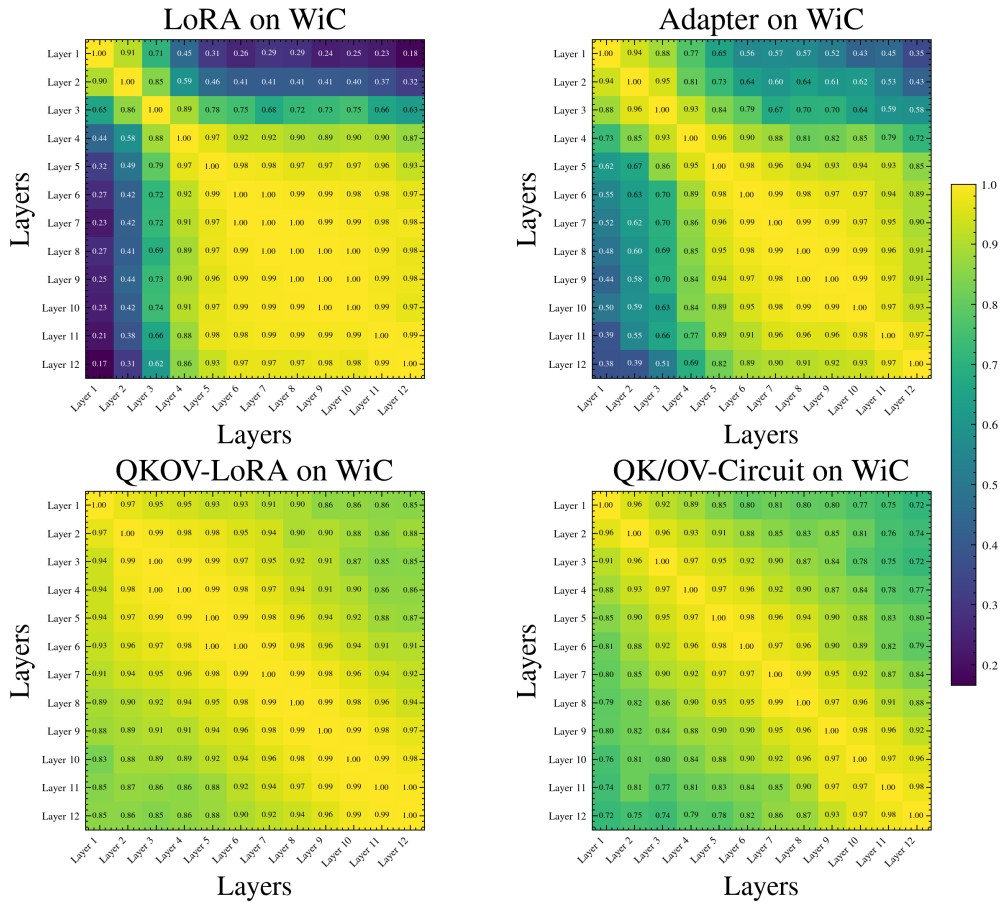

Figure 4: Layer-wise Linear CKA similarity on **WiC** (drop case). QKOV-LoRA has CKA 0.85 between Layer 1 and Layer 12, which is much higher than on CoLA (0.55), indicating weaker depth-wise differentiation on WiC. QK/OV-Circuit shows the same trend (WiC: 0.72 vs. CoLA: 0.29). LoRA and Adapter preserve a similar two-phase pattern across tasks, while depth-dependent methods exhibit task-dependent depth profiles.

### 5.3.2 Token cosine similarity

To complement CKA, we examine within-layer geometry using the distribution of pairwise cosine similarities among token vectors. Figures 5 and 6 show four plots: LoRA vs. QKOV-LoRA on a gain case (CoLA) and a drop case (WiC).

**CoLA.** LoRA shows broad and multimodal distributions across all 12 layers, with substantial mass between approximately $-0.5$ and $1.0$. Even in late layers (9–12), multiple peaks remain, suggesting that angular diversity between tokens is preserved. In contrast, QKOV-LoRA shows a rapid collapse toward cosine similarity $\approx 1.0$ by layers 3–4, and an almost delta-like mass near $1.0$ after layer 5. This pattern indicates that QKOV-LoRA induces strong within-layer alignment of token representations.

**WiC.** LoRA again shows a broad and multimodal distribution, similar to CoLA, and the task dependence is small. QKOV-LoRA shows the same rapid collapse toward cosine similarity $\approx 1.0$ as in CoLA. This indicates that QKOV-LoRA induces token aggregation in both the gain and drop cases; token cosine similarity alone therefore does not, by itself, account for the task-dependent outcome. CKA provides a difference between these two tasks: on CoLA, token aggregation coincides with stronger depth-wise differentiation (lower Layer 1–12 CKA), while on WiC it coincides with weaker differentiation (higher Layer 1–12 CKA). In this comparison, the gain case coincides with token aggregation together with a depth profile that changes across layers, rather than token aggregation alone.

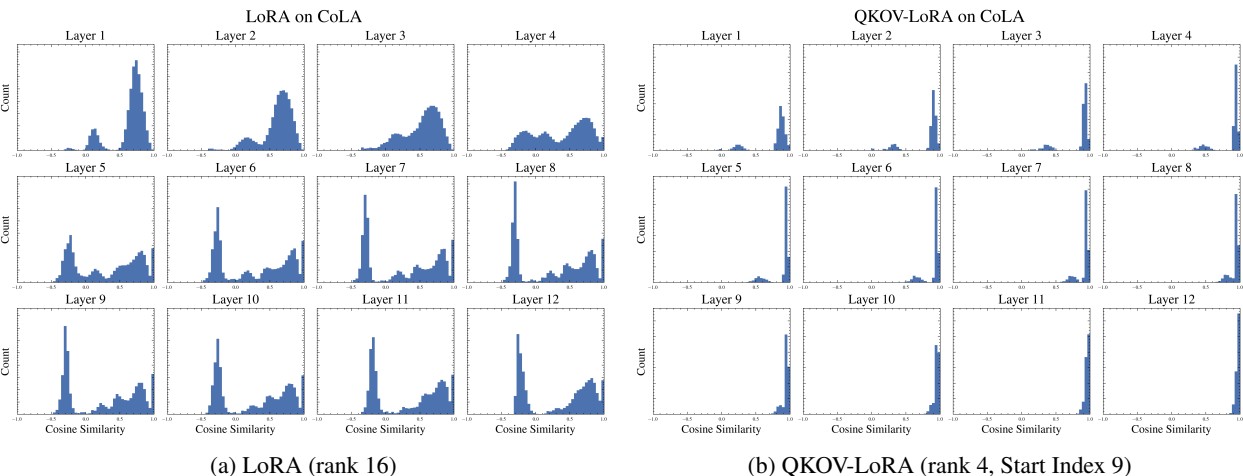

(a) LoRA (rank 16)        (b) QKOV-LoRA (rank 4, Start Index 9)

Figure 5: Token cosine similarity distributions on **CoLA** (gain case). Under LoRA (a), the distributions remain broad (approximately $-0.5$ to $1.0$) and multimodal across layers. Under QKOV-LoRA (b), the distributions rapidly concentrate near cosine similarity $\approx 1.0$ by layers 3–4 and collapse to an almost delta-like mass near $1.0$ after layer 5.

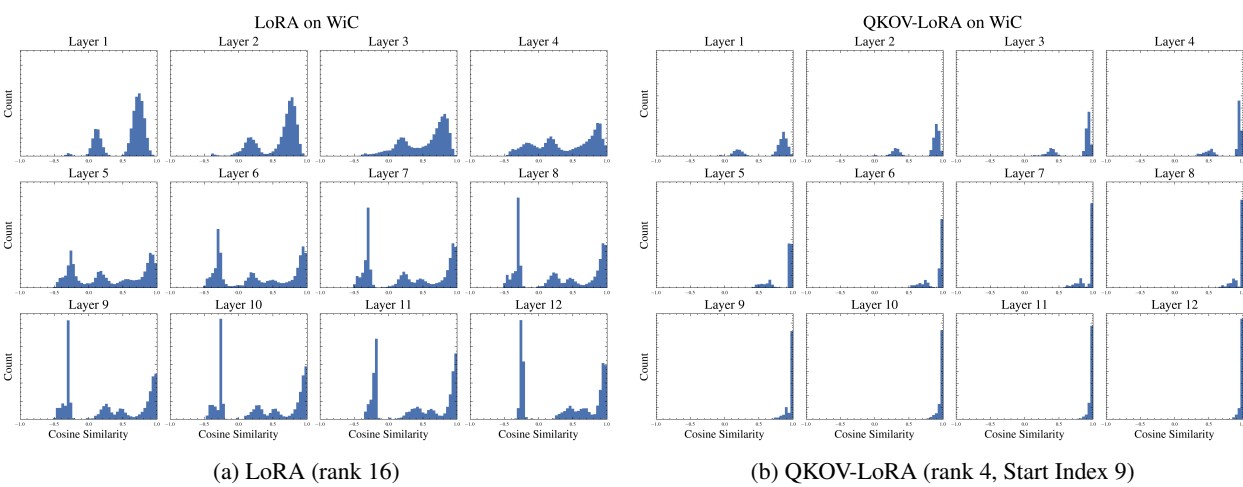

(a) LoRA (rank 16)        (b) QKOV-LoRA (rank 4, Start Index 9)

Figure 6: Token cosine similarity distributions on **WiC** (drop case). Under LoRA (a), the distributions remain broad and multimodal across layers, similar to CoLA. Under QKOV-LoRA (b), the same rapid concentration toward cosine similarity $\approx 1.0$ is observed, indicating that token aggregation alone is not sufficient to distinguish gain and drop cases.

## 5.4 Results on Computer Vision

### 5.4.1 Component-Wise Sharing Analysis

MHSA governs inter-token interactions and may require different behavior at different depths, while FFN applies a per-token transformation that is more amenable to sharing. To test this, we construct three ViT variants with recurrence beginning at a specified Start Index (the first recurrent layer, i.e., layer $l + 1$): (i) sharing the **Full** block, (ii) sharing only **MHSA** (unique FFNs), and (iii) sharing only **FFN** (unique MHSAs).

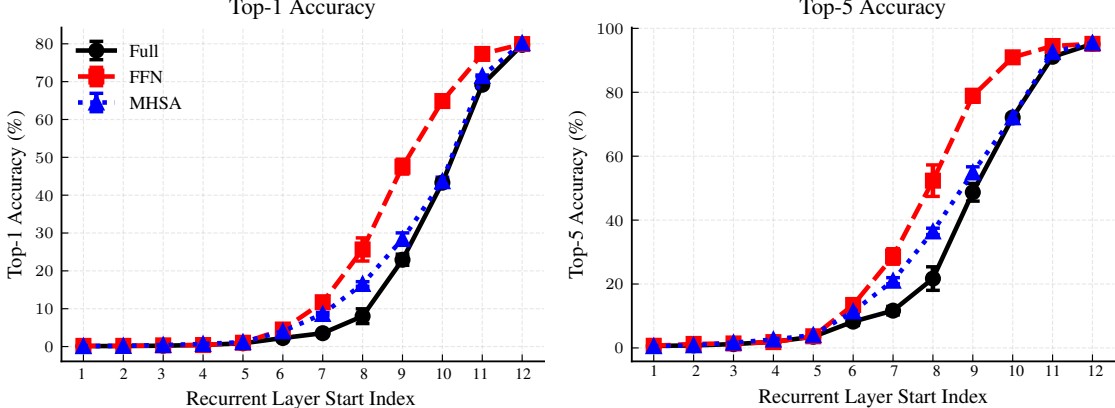

Figure 7: **Component-wise parameter sharing in ViT-B/16 on ImageNet.** Top-1 and Top-5 accuracy vs. Start Index (the first recurrent layer). Single-crop, center evaluation.

As shown in Figure 7, sharing MHSA yields a large drop in accuracy compared to sharing FFN. For example, when sharing begins at the 8th layer, the FFN-sharing model maintains Top-1 above 40%, whereas the MHSA-sharing model drops below 20%. This pattern is consistent with MHSA being more depth-sensitive than FFN: earlier layers tend to establish local relations while later layers integrate global context. The FFN transformation appears more uniform across depth and may be more amenable to sharing. This analysis motivates our choice to modulate MHSA while keeping FFN fixed.

### 5.4.2 ImageNet Evaluation with Matched Parameters

We evaluate QKOV-LoRA and QK/OV-Circuit on ImageNet-1k alongside three categories of baselines: (a) pure weight sharing with no adaptation, (b) depth-invariant LoRA applied to the shared block at every recurrence step, and (c) MiniViT (Zhang et al., 2022), a distillation-based compression method. To reduce confounding from parameter budget, we match the number of trainable parameters as closely as possible across methods.

Figure 8 compares QKOV-LoRA and QK/OV-Circuit (rank 16) against pure sharing and depth-invariant LoRA across Start Index values. In this figure only, rank is fixed to 16 for all LoRA-based methods so that the sweep isolates the effect of Start Index. Depth-invariant baselines recover little accuracy over naive sharing: at Start Index 7, the depth-invariant LoRA reaches approximately 40% Top-1 while both proposed methods reach above 60%. This gap persists across the full range of Start Index values.

Table 4 reports a matched-parameter comparison to more precisely isolate the effect of depth-varying modulation. Methods are grouped into two parameter bands (approximately 123k and 98.3k trainable parameters). Within each band, we include the following baselines. LoRA FFN (rank 16) applies a shared LoRA module to the FFN block at every recurrence step, and LoRA MHSA (rank 16) applies a shared LoRA module to the MHSA block. Both are depth-invariant: the LoRA parameters do not change with layer index, and their total parameter count is independent of the Start Index. The Start Index values in the table for these baselines match those of the proposed methods to keep the number of recurrent layers constant. MiniViT compresses Vision Transformers through weight multiplexing, training a compact student model by knowledge distillation from the same pretrained ViT-B/16 that serves as the backbone in our approach. The Mini-DeiT-Ti variants used here are configured to match each parameter band; specifications are in Appendix A.4.

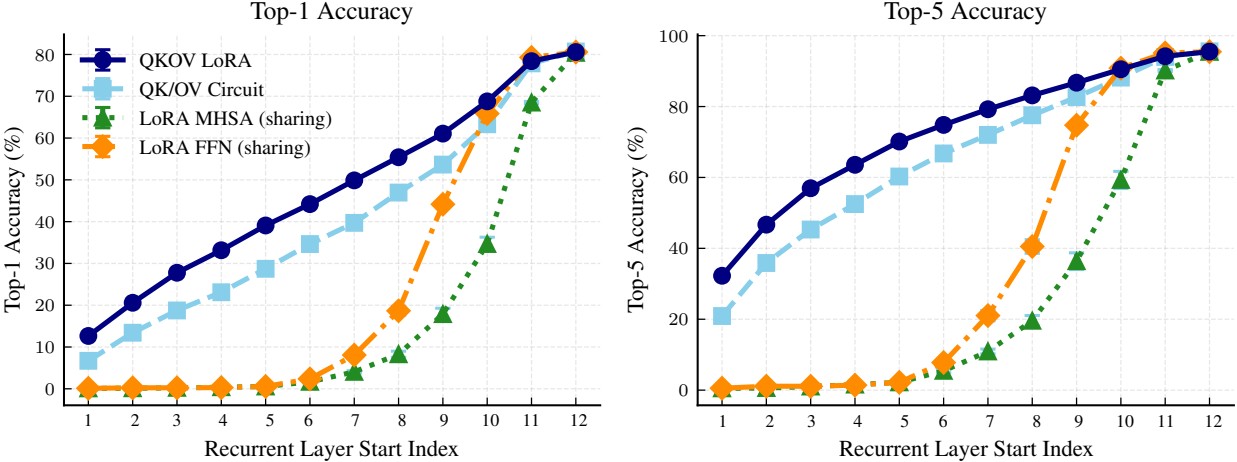

Figure 8: **QKOV-LoRA and QK/OV-Circuit on ImageNet.** Top-1 and Top-5 accuracy (rank 16) vs. (i) pure sharing and (ii) depth-invariant LoRA. This figure is a fixed-rank sweep (rank 16 for both proposed methods and the depth-invariant LoRA baseline) over Start Index values. For matched-parameter comparisons with rank variation, see Table 4.

Table 4: ImageNet-1k results with 100% training data, grouped by matched trainable-parameter budget (upper: ≈123k, lower: ≈98.3k). Accuracy: mean ± std over 5 seeds.

| Method | Rank | Start Index | Params (k) | GFLOPs | Top-1 Acc. (%) | Top-5 Acc. (%) |
|---|---|---|---|---|---|---|
| LoRA FFN | 16 | 7 | 123 | 35.4 | 25.7 ± 3.1 | 52.3 ± 4.9 |
| LoRA FFN | 16 | 8 | 123 | 35.3 | 47.5 ± 2.0 | 78.9 ± 2.0 |
| MiniViT (Mini-DeiT-Ti) | - | - | 123 | 0.0989 | 22.4 ± 0.24 | 44.5 ± 0.44 |
| **QKOV-LoRA** | 4 | 7 | 123 | 35.2 | 61.5 ± 0.062 | 86.3 ± 0.12 |
| **QK/OV-Circuit** | 15 | 7 | 125 | 40.1 | **63.5 ± 0.061** | **87.4 ± 0.052** |
| LoRA MHSA | 16 | 7 | 98.3 | 35.3 | 16.5 ± 0.61 | 36.5 ± 1.0 |
| LoRA MHSA | 16 | 8 | 98.3 | 35.3 | 28.4 ± 1.6 | 54.9 ± 1.8 |
| MiniViT (Mini-DeiT-Ti) | - | - | 98.3 | 0.0398 | 15.2 ± 0.16 | 33.3 ± 0.39 |
| **QKOV-LoRA** | 4 | 8 | 98.3 | 35.2 | 65.6 ± 0.16 | 88.8 ± 0.098 |
| **QK/OV-Circuit** | 15 | 8 | 99.8 | 39.1 | **66.9 ± 0.093** | **89.4 ± 0.063** |

In the 123k band, QK/OV-Circuit achieves 63.5±0.061% Top-1 and 87.4±0.052% Top-5, followed by QKOV-LoRA at 61.5±0.062% and 86.3±0.12%. LoRA FFN at Start Index 8 reaches 47.5±2.0% Top-1, and MiniViT reaches 22.4±0.24%. In the 98.3k band, the comparison between LoRA MHSA and QKOV-LoRA helps isolate the effect of depth-varying modulation, because both adapt the MHSA component with matched parameters: LoRA MHSA uses rank 16 shared across layers, while QKOV-LoRA uses rank 4 with layer-specific modules. At Start Index 8, QKOV-LoRA reaches 65.6±0.16% Top-1 compared to 28.4±1.6% for LoRA MHSA, a difference of 37.2 points. QK/OV-Circuit achieves 66.9±0.093%, and MiniViT reaches 15.2±0.16%.

In computational cost, QKOV-LoRA has GFLOPs comparable to the LoRA baselines (approximately 35 GFLOPs), while QK/OV-Circuit requires somewhat more computation (39–40 GFLOPs) due to the composite QK and OV operations (Section 4). MiniViT has lower GFLOPs (below 0.1) because distillation produces a compact model with fewer layers and smaller hidden dimensions. The accuracy gap between the proposed methods and the LoRA baselines is wider for MHSA-based adaptation (lower band) than for FFN-based adaptation (upper band), consistent with the component-wise analysis in Section 5.4.1.

Table 5: ImageNet-1k results with 10% training data. Rows are paired by matched trainable parameters: MiniViT (Mini-DeiT-Ti variant) above, proposed method below. Accuracy: mean ± std over 5 seeds.

| Method | Rank | Start Index | Top-1 Acc. (%) | Params (M) | GFLOPs |
|---|---|---|---|---|---|
| ViT-B/16(Base) | - | - | 84.53 | 86.7 | 17.56 |
| MiniViT (Mini-DeiT-Ti) | - | - | 11.27 ± 0.26 | 1.08 | 1.201 |
| QKOV-LoRA | 16 | 1 | 18.48 ± 4.60 | 1.08 | 35.58 |
| MiniViT (Mini-DeiT-Ti) | - | - | 10.51 ± 0.37 | 0.785 | 0.6887 |
| QKOV-LoRA | 16 | 4 | 40.61 ± 1.58 | 0.786 | 35.46 |
| MiniViT (Mini-DeiT-Ti) | - | - | 9.46 ± 0.25 | 0.492 | 0.3404 |
| QKOV-LoRA | 16 | 7 | 57.83 ± 0.20 | 0.491 | 35.34 |
| MiniViT (Mini-DeiT-Ti) | - | - | 7.52 ± 0.081 | 0.197 | 0.1071 |
| QKOV-LoRA | 16 | 10 | 79.52 ± 0.10 | 0.197 | 35.22 |
| MiniViT (Mini-DeiT-Ti) | - | - | 8.19 ± 0.2 | 0.293 | 0.3126 |
| QK/OV-Circuit | 16 | 1 | 21.73 ± 0.21 | 0.293 | 46.02 |
| MiniViT (Mini-DeiT-Ti) | - | - | 7.64 ± 0.27 | 0.213 | 0.2253 |
| QK/OV-Circuit | 16 | 4 | 41.52 ± 0.07 | 0.213 | 43.05 |
| MiniViT (Mini-DeiT-Ti) | - | - | 7.02 ± 0.32 | 0.133 | 0.1166 |
| QK/OV-Circuit | 16 | 7 | 57.75 ± 0.17 | 0.133 | 40.08 |
| MiniViT (Mini-DeiT-Ti) | - | - | 3.72 ± 0.13 | 0.0532 | 0.02146 |
| QK/OV-Circuit | 16 | 10 | 79.46 ± 0.03 | 0.0532 | 37.12 |

### 5.4.3 Performance in a Low-Data Regime

We also evaluate in a low-data regime using only 10% of the ImageNet-1k training set. This setting tests whether the modulated models generalize when training data is limited, a condition where overfitting to limited supervision is a concern. Because the proposed methods keep the pretrained ViT-B/16 backbone frozen and train only small depth-indexed modules, the pretrained features are largely preserved. Table 5 compares QKOV-LoRA and QK/OV-Circuit against MiniViT at matched trainable-parameter budgets across four Start Index values (1, 4, 7, 10).

At all Start Index values, both proposed methods are higher than the matched MiniViT baselines. For example, at Start Index 10 with approximately 0.197M parameters, QKOV-LoRA reaches 79.52±0.10% Top-1 while the matched MiniViT reaches 7.52±0.081%; at Start Index 1 with approximately 0.293M parameters, QK/OV-Circuit reaches 21.73±0.21% while MiniViT reaches 8.19±0.2%. For the proposed methods, accuracy increases with the Start Index even though the trainable-parameter budget decreases. This trend may reflect that a higher Start Index preserves more

pretrained layers, so the frozen pretrained features provide a stronger initialization when modulation capacity is reduced. In this setup, MiniViT shows the opposite trend, with lower accuracy at smaller parameter budgets, consistent with reduced model capacity under weight multiplexing.

Comparing the two proposed methods, QK/OV-Circuit achieves higher accuracy than QKOV-LoRA at Start Index 1 and 4 (21.73% vs 18.48% and 41.52% vs 40.61% respectively), while both methods converge to similar accuracy at Start Index 7 and 10 (57.75% vs 57.83% and 79.46% vs 79.52%).

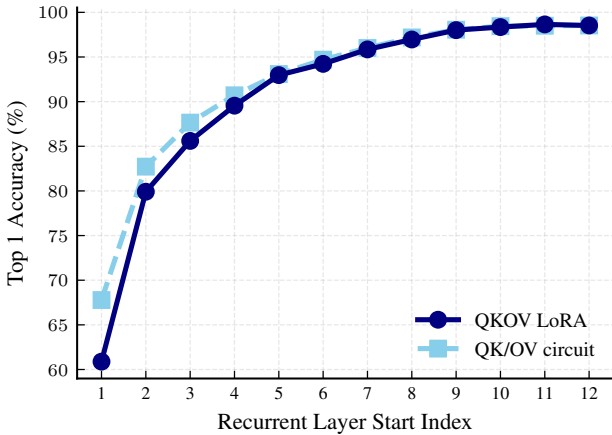

Figure 9: **Downstream performance on CIFAR-10.** Top-1 accuracy after fine-tuning on CIFAR-10 as a function of Start Index.

### 5.4.4 Performance on a Downstream Classification Task

We fine-tune the ImageNet-trained models on CIFAR-10 to evaluate downstream transfer. Figure 9 reports Top-1 accuracy as a function of Start Index. Both QKOV-LoRA and QK/OV-Circuit show increasing CIFAR-10 accuracy as the Start Index increases, consistent with preserving more pretrained layers providing a stronger feature backbone for transfer.

QK/OV-Circuit achieves slightly higher accuracy than QKOV-LoRA at lower Start Index values, where more layers use the recurrent structure. At higher Start Index values, both methods achieve comparable accuracy. This pattern is consistent with the ImageNet and low-data results.

## 6 Conclusion

We examined depth-indexed low-rank modulation as a lightweight way to relax the uniformity of weight-sharing Transformers. By freezing the shared block and training only small modules inside multi-head self-attention (MHSA) (while keeping the feed-forward network unchanged), the transformation can vary across depth while most parameters remain fixed.

We instantiated this idea in two forms: QKOV-LoRA, which applies depth-indexed Low-Rank Adaptation (LoRA) to the $Q, K, V, O$ projections, and QK/OV-Circuit, which applies low-rank corrections to the QK routing and OV aggregation operators, trading flexibility for a more constrained parameterization with different compute characteristics.

Across our matched-budget comparisons, depth-indexed MHSA modulation is generally effective in the vision settings we studied, including low-data regimes. In language, the impact is mixed and varies by task, so depth variation can help some tasks while hurting others.

Several limitations remain. Choosing where modulation starts (Start Index) is currently a task- and architecture-dependent hyperparameter that we tune by sweeps. Our design targets MHSA and does not directly extend to non-attention recurrent architectures such as RWKV (Peng et al., 2023) or Mamba (Gu & Dao, 2024). In QK/OV-Circuit, the learned correction factors depend on the token count, which ties trained weights to the sequence length used during training and complicates transfer to different lengths.

Future work includes methods to select where and how much to modulate more automatically, and extensions of the modulation interface beyond MHSA-based Transformers.

## Broader Impact Statement

We study parameter-efficient adaptation of weight-sharing Transformers via the depth-indexed MHSA modulation approach described in Section 4. By reducing the number of trainable parameters required for adaptation, the approach may reduce compute and memory requirements for fine-tuning and deployment in resource-constrained environments. Compression and parameter sharing can also change performance unevenly across downstream tasks or domains; deployment should therefore be preceded by evaluation on the intended setting and appropriate monitoring.

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

## A    Experimental Specifications

This appendix summarizes the experimental configurations and hyperparameters used in this work, complementing the descriptions in the main text. All experiments were implemented in PyTorch (Paszke et al., 2019) and the Hugging Face Transformers library (Wolf et al., 2020), and executed on NVIDIA RTX 6000 Ada Generation GPUs.

### A.1    Language Experiments (GLUE and SuperGLUE)

We evaluate ALBERT-base-v2 (Lan et al., 2020) on GLUE (Wang et al., 2018) and SuperGLUE (Wang et al., 2019). For all language results, we report mean and standard deviation over 10 random seeds. Unless otherwise noted, we optimize with AdamW (Loshchilov & Hutter, 2019) and use a cosine learning-rate schedule with warmup, and we clip the gradient norm at 1.0.

We use Start Index to denote the first layer where depth-indexed modulation is activated; layers before the Start Index reuse the shared ALBERT block without depth-indexed modulation.

**GLUE task schedules.**

- **CoLA:** 10 epochs, batch size 16, max sequence length 128.

- **SST-2:** 3 epochs, batch size 32, max sequence length 128.

- **MRPC:** 10 epochs, batch size 16, max sequence length 256.

- **STS-B:** 10 epochs, batch size 16, max sequence length 256.

- **RTE:** 10 epochs, batch size 16, max sequence length 256.

- **QQP:** 3 epochs, batch size 32, max sequence length 256.

- **MNLI:** 3 epochs, batch size 32, max sequence length 256.

- **QNLI:** 3 epochs, batch size 32, max sequence length 256.

**SuperGLUE task schedules.**

- **BoolQ & WiC:** 24 epochs, batch size 16, max sequence length 256.

- **CB:** 48 epochs, batch size 8, max sequence length 256.

- **COPA:** 72 epochs, batch size 8, max sequence length 160.

- **MultiRC:** 16 epochs, batch size 8, max sequence length 512.

- **RTE:** 40 epochs, batch size 12, max sequence length 256.

- **WSC:** 80 epochs, batch size 8, max sequence length 384.

**Additional Start Index sweep on SuperGLUE.** Table 6 reports an additional Start Index sweep on the SuperGLUE development set. Full finetuning and LoRA do not use Start Index, while the proposed methods vary Start Index from 1 to 12. Values in parentheses denote the absolute difference from the LoRA baseline within each column; this table is included as a supplementary sweep in addition to the matched-parameter comparisons in the main text. We additionally include a LoRA baseline at rank 4 as a rank-matched reference point for QKOV-LoRA (rank 4). On the SuperGLUE development set, LoRA (rank 4) averages 64.1 across tasks, 3.8 points below LoRA (rank 16). At the Start Index values with the highest averages in Table 6, QKOV-LoRA (Start Index 8, average 69.1) and QK/OV-Circuit (Start Index 6, average 68.9) are higher than the rank-4 LoRA baseline by 5.0 and 4.8 points respectively. This sweep provides a rank-matched reference point for interpreting depth-indexed modulation at lower rank in this setting.

## A.2 ImageNet Experiments

**Base Model Specifications.** The base model for all vision experiments was the standard Vision Transformer (ViT-B/16) pretrained on ImageNet-21k and fine-tuned on ImageNet-1k, as described by Dosovitskiy et al. (2021). This model has 12 layers ($L = 12$), a hidden dimension of 768, 12 attention heads, and processes images as a sequence of 16x16 patches.

**Dataset and Preprocessing.** We conducted large-scale experiments on the ImageNet-1k dataset (Deng et al., 2009). For the low-data regime experiments, models were trained on a randomly selected 10% subset of the full ImageNet-1k training data. All models were evaluated on the standard, full ImageNet-1k validation set. Input images were resized to 224x224 pixels. Pixel values were normalized using the standard ImageNet mean and standard deviation.

Table 6: SuperGLUE development set results. We report mean ± standard deviation over 10 random seeds. Values in parentheses denote absolute improvement over the LoRA (rank 16) baseline. Start Index denotes the first layer where depth-indexed modulation is activated.

| Method | Start Index | BoolQ | CB | COPA | MultiRC | RTE | WiC | WSC | Ave. |
|---|---|---|---|---|---|---|---|---|---|
| **Full finetuning** | – | 77.0±1.0 | 83.9±3.9 | 67.7±1.2 | 78.8±0.2 | 75.5±1.8 | 71.0±0.7 | 59.9±2.8 | 73.4 |
| **LoRA (rank 16)** | – | 70.1±1.0 | 66.7±0.8 | 61.7±4.6 | 77.4±0.4 | 70.6±2.5 | 70.5±1.2 | 58.0±3.3 | 67.9 |
| **LoRA (rank 4)** | – | 68.9±0.8 | 67.7±1.9 | 47.5±4.0 | 75.6±0.7 | 65.7±2.3 | 65.0±1.0 | 58.1±3.0 | 64.1 |
| **QKOV-LoRA (rank 4)** | 1 | 73.1±0.8 *(+3.1)* | 74.4±4.5 *(+7.7)* | 62.0±1.4 *(+0.3)* | 76.2±0.1 *(-1.2)* | 70.6±1.0 *(0.0)* | 69.3±0.5 *(-1.3)* | 44.9±2.0 *(-13.1)* | 67.2 *(-0.6)* |
| **QKOV-LoRA (rank 4)** | 2 | 72.9±0.5 *(+2.8)* | 74.4±2.2 *(+7.7)* | 61.0±1.6 *(-0.7)* | 76.3±0.8 *(-1.1)* | 71.1±0.5 *(+0.5)* | 68.8±1.4 *(-1.8)* | 49.4±3.0 *(-8.7)* | 67.7 *(-0.2)* |
| **QKOV-LoRA (rank 4)** | 3 | 73.4±0.3 *(+3.3)* | 76.2±3.0 *(+9.5)* | 65.3±1.9 *(+3.7)* | 75.8±0.3 *(-1.6)* | 68.6±1.1 *(-2.0)* | 67.3±0.7 *(-3.2)* | 48.7±5.5 *(-9.3)* | 67.9 *(+0.1)* |
| **QKOV-LoRA (rank 4)** | 4 | 72.3±0.4 *(+2.3)* | 77.4±5.9 *(+10.7)* | 64.0±2.4 *(+2.3)* | 74.0±2.8 *(-3.4)* | 72.6±3.1 *(+2.0)* | 69.3±1.4 *(-1.2)* | 51.9±0.8 *(-6.1)* | 68.8 *(+0.9)* |
| **QKOV-LoRA (rank 4)** | 5 | 72.8±0.6 *(+0.9)* | 78.0±2.2 *(+11.3)* | 64.7±2.1 *(+3.0)* | 75.7±0.4 *(-1.7)* | 70.8±1.6 *(+0.1)* | 68.3±0.7 *(-2.2)* | 50.3±3.9 *(-7.7)* | 68.7 *(+0.8)* |
| **QKOV-LoRA (rank 4)** | 6 | 69.3±0.3 *(-0.8)* | 70.8±0.8 *(+4.1)* | 60.0±3.6 *(-1.7)* | 75.4±0.2 *(-2.0)* | 69.2±1.4 *(-1.5)* | 66.7±1.0 *(-3.8)* | 54.5±1.8 *(-3.5)* | 66.6 *(-1.3)* |
| **QKOV-LoRA (rank 4)** | 7 | 73.2±0.7 *(+3.1)* | 82.7±6.7 *(+16.0)* | 63.7±3.7 *(+2.0)* | 75.3±0.3 *(-2.1)* | 71.6±1.3 *(+1.0)* | 68.1±1.7 *(-2.4)* | 47.1±0.8 *(-10.9)* | 68.8 *(+1.0)* |
| **QKOV-LoRA (rank 4)** | 8 | 72.9±0.8 *(+2.8)* | 76.8±2.9 *(+10.1)* | 64.0±2.9 *(+2.3)* | 75.0±0.3 *(-2.4)* | 72.3±2.1 *(+1.6)* | 67.2±0.3 *(-3.3)* | 55.4±3.3 *(-2.6)* | 69.1 *(+1.2)* |
| **QKOV-LoRA (rank 4)** | 9 | 72.6±0.1 *(+2.5)* | 81.0±2.2 *(+14.3)* | 63.0±1.6 *(+1.3)* | 74.3±0.5 *(-3.1)* | 72.0±1.5 *(+1.3)* | 66.1±1.1 *(-4.4)* | 53.8±2.8 *(-4.2)* | 69.0 *(+1.1)* |
| **QKOV-LoRA (rank 4)** | 10 | 72.1±0.3 *(+2.1)* | 81.5±6.1 *(+14.9)* | 62.3±0.9 *(+0.7)* | 73.5±0.3 *(-3.9)* | 69.7±1.5 *(-1.0)* | 67.4±1.6 *(-3.1)* | 51.0±3.4 *(-7.0)* | 68.2 *(+0.4)* |
| **QKOV-LoRA (rank 4)** | 11 | 72.0±0.4 *(+2.0)* | 81.5±7.2 *(+14.9)* | 62.7±2.6 *(+1.0)* | 71.8±0.6 *(-5.6)* | 70.9±1.2 *(+0.2)* | 66.6±0.4 *(-4.0)* | 52.2±1.2 *(-5.8)* | 68.2 *(+0.4)* |
| **QKOV-LoRA (rank 4)** | 12 | 70.3±0.7 *(+0.2)* | 76.8±0.0 *(+10.1)* | 62.7±2.1 *(+1.0)* | 71.1±0.1 *(-6.3)* | 67.5±1.9 *(-3.1)* | 64.3±1.7 *(-6.2)* | 59.0±3.7 *(+1.0)* | 67.4 *(-0.5)* |
| **QK/OV-Circuit (rank 4)** | 1 | 69.3±0.7 *(-0.7)* | 72.0±2.2 *(+5.4)* | 58.3±1.2 *(-3.3)* | 76.0±0.8 *(-1.4)* | 66.5±2.7 *(-4.1)* | 68.4±1.3 *(-2.1)* | 51.6±4.3 *(-6.4)* | 66.0 *(-1.8)* |
| **QK/OV-Circuit (rank 4)** | 2 | 69.8±0.5 *(-0.3)* | 73.8±2.2 *(+7.1)* | 58.0±2.2 *(-3.7)* | 76.0±0.2 *(-1.4)* | 66.3±2.1 *(-4.3)* | 68.3±0.6 *(-2.2)* | 50.6±1.6 *(-7.4)* | 66.1 *(-1.7)* |
| **QK/OV-Circuit (rank 4)** | 3 | 70.2±1.0 *(+0.1)* | 78.0±3.0 *(+11.3)* | 62.0±2.2 *(+0.3)* | 76.2±0.7 *(-1.2)* | 65.9±2.7 *(-4.8)* | 67.8±0.8 *(-2.7)* | 51.3±1.6 *(-6.7)* | 67.4 *(-0.5)* |
| **QK/OV-Circuit (rank 4)** | 4 | 69.9±0.7 *(-0.2)* | 72.6±3.0 *(+5.9)* | 61.0±2.8 *(-0.7)* | 76.0±0.4 *(-1.4)* | 66.8±2.6 *(-3.8)* | 68.2±1.0 *(-2.3)* | 52.2±0.5 *(-5.8)* | 66.7 *(-1.2)* |
| **QK/OV-Circuit (rank 4)** | 5 | 71.0±0.1 *(+0.9)* | 72.0±1.7 *(+5.3)* | 61.7±2.6 *(0.0)* | 76.4±0.5 *(-1.0)* | 68.4±2.1 *(-2.2)* | 65.8±1.1 *(-4.8)* | 51.6±3.7 *(-6.4)* | 66.7 *(-1.2)* |
| **QK/OV-Circuit (rank 4)** | 6 | 70.5±0.3 *(+0.4)* | 76.2±2.2 *(+9.5)* | 65.7±2.6 *(+4.0)* | 75.7±0.2 *(-1.7)* | 68.6±1.6 *(-2.0)* | 68.9±0.9 *(-1.6)* | 57.1±3.3 *(-1.0)* | 68.9 *(+1.1)* |
| **QK/OV-Circuit (rank 4)** | 7 | 70.4±0.1 *(+0.3)* | 69.0±2.2 *(+2.3)* | 63.7±1.7 *(+2.0)* | 75.3±0.2 *(-2.1)* | 69.2±2.5 *(-1.5)* | 69.3±0.6 *(-1.2)* | 54.2±4.3 *(-3.8)* | 67.3 *(-0.6)* |
| **QK/OV-Circuit (rank 4)** | 8 | 70.4±0.4 *(+0.3)* | 72.0±3.4 *(+5.3)* | 64.3±1.7 *(+2.6)* | 74.4±0.8 *(-3.0)* | 71.8±1.8 *(+1.1)* | 69.2±0.7 *(-1.3)* | 58.7±2.1 *(+0.6)* | 68.7 *(+0.8)* |
| **QK/OV-Circuit (rank 4)** | 9 | 70.2±0.7 *(+0.1)* | 73.8±3.7 *(+7.1)* | 65.7±3.3 *(+4.0)* | 73.3±0.3 *(-4.1)* | 72.8±2.5 *(+2.2)* | 67.7±1.5 *(-2.9)* | 57.1±1.8 *(-1.0)* | 68.6 *(+0.8)* |
| **QK/OV-Circuit (rank 4)** | 10 | 69.4±0.2 *(-0.7)* | 73.2±1.5 *(+6.5)* | 65.7±3.3 *(+4.0)* | 71.6±0.2 *(-5.8)* | 69.7±1.4 *(-1.0)* | 67.0±1.3 *(-3.5)* | 58.0±2.4 *(0.0)* | 67.8 *(-0.1)* |
| **QK/OV-Circuit (rank 4)** | 11 | 66.9±0.1 *(-3.2)* | 68.5±0.8 *(+1.9)* | 61.0±2.8 *(-0.7)* | 70.0±0.3 *(-7.4)* | 67.5±1.1 *(-3.1)* | 65.4±0.6 *(-5.1)* | 59.3±2.5 *(+1.3)* | 65.5 *(-2.3)* |
| **QK/OV-Circuit (rank 4)** | 12 | 65.0±0.5 *(-5.0)* | 65.5±2.2 *(-1.2)* | 55.3±1.2 *(-6.3)* | 68.3±0.0 *(-9.1)* | 66.3±0.3 *(-4.3)* | 63.4±0.7 *(-7.1)* | 60.6±0.0 *(+2.6)* | 63.5 *(-4.4)* |

**Training and Optimization.** All models were trained for 100 epochs using a total batch size of 1024 (distributed across GPUs). For the Start Index sweep in Figure 8 and the 10% data setting in Table 5, we set the LoRA rank to 16 for both QKOV-LoRA and QK/OV-Circuit. For the 100% matched-parameter comparison in Table 4, ranks follow the main-table settings (e.g., QKOV-LoRA rank 4 and QK/OV-Circuit rank 15 in the reported runs). The models were trained using the AdamW optimizer with a base learning rate of $1 \times 10^{-3}$, which was scaled linearly with the batch size. We used a weight decay of 0.05 and a cosine learning rate scheduler with a 5-epoch linear warmup. The loss function was the standard cross entropy loss with label smoothing (smoothing factor of 0.1). To account for training variability, all reported ImageNet accuracies are the mean and standard deviation calculated over 5 random seeds.

## A.3 CIFAR-10 Downstream Task Experiments

**Fine-tuning Setup.** To evaluate transfer learning performance, the ImageNet-trained models (with their respective compressed structures and learned depth-indexed modules) were fine-tuned on the CIFAR-10 dataset (Krizhevsky et al., 2009). This process assesses the generalizability of the learned representations.

**Data Preprocessing and Augmentation.** Input images from CIFAR-10 were resized to 224x224 to match the ViT's expected input resolution. During training, we applied standard data augmentations for CIFAR-10, including random cropping (with padding) and random horizontal flipping. During evaluation, only a center crop was used. Images were normalized using the CIFAR-10 dataset's mean and standard deviation.

**Optimization.** The models were fine-tuned for 100 epochs with a batch size of 64. We used the AdamW optimizer with a learning rate of $1 \times 10^{-4}$ and a weight decay of $1 \times 10^{-4}$. Gradients were clipped at a maximum norm of 1.0. The learning rate was managed by a cosine annealing scheduler with a 500-step linear warmup phase to ensure stable fine-tuning. The loss function was Cross-Entropy Loss.

## A.4 MiniViT (Mini-DeiT-Ti) Baseline Details

**Baseline description.** We compare against MiniViT-style baselines derived from Mini-DeiT-Ti (Zhang et al., 2022). MiniViT is a compression framework for Vision Transformers that reduces parameters by multiplexing weights across consecutive blocks and, when used in the original recipe, can incorporate distillation. Mini-DeiT(-Ti) denotes a family of compact DeiT-style Vision Transformers produced within this framework (Zhang et al., 2022; Touvron et al., 2021a).

In Tables 7 and 8, we denote the matched Mini-DeiT variants as Mini-DeiT (Method (Start Index)), where Method indicates which proposed method the configuration matches (QKOV-LoRA or QK/OV-Circuit), and the number indicates the Start Index used in the corresponding comparison (1, 4, 7, or 10). The remaining variants are labeled by the baseline they match in the full-data parameter-band comparisons. Across both tables, Params denotes the number of model parameters, Depth the number of transformer blocks, Embed Dim the embedding dimension, Heads the number of attention heads, and MLP Ratio the feed-forward expansion ratio.

**Full-data (ImageNet 100%) setting.** For the full training data comparisons, we use smaller Mini-DeiT-Ti variants tuned to match the trainable-parameter bands reported in the main tables.

Table 7: Mini-DeiT-Ti variants used for the ImageNet 100% training data comparisons.

| Model | Params | Depth | Embed Dim | Heads | MLP Ratio |
|---|---|---|---|---|---|
| Mini-DeiT (LoRA FFN) | 123012 | 6 | 40 | 4 | 2 |
| Mini-DeiT (LoRA MHSA) | 98296 | 2 | 40 | 2 | 3 |
| Mini-DeiT-Ti Patch16-224 (original) | 3090304 | 12 | 192 | 3 | 4 |

**Low-data (ImageNet 10%) setting.** For the 10% training data regime, we evaluated additional compressed variants (further reduced from the Mini-DeiT-Ti family) with the following configurations.

Table 8: Mini-DeiT-Ti variants used for the ImageNet 10% training data comparisons.

| Model | Params | Depth | Embed Dim | Heads | MLP Ratio |
|---|---|---|---|---|---|
| Mini-DeiT (QKOV-LoRA (1)) | 1081120 | 20 | 80 | 8 | 5 |
| Mini-DeiT (QKOV-LoRA (4)) | 784744 | 12 | 72 | 4 | 8 |
| Mini-DeiT (QKOV-LoRA (7)) | 491992 | 4 | 88 | 4 | 8 |
| Mini-DeiT (QKOV-LoRA (10)) | 196754 | 2 | 72 | 8 | 3 |
| Mini-DeiT (QK/OV-Circuit (1)) | 292788 | 16 | 36 | 4 | 8 |
| Mini-DeiT (QK/OV-Circuit (4)) | 213066 | 18 | 34 | 2 | 4 |
| Mini-DeiT (QK/OV-Circuit (7)) | 133140 | 10 | 30 | 2 | 5 |
| Mini-DeiT (QK/OV-Circuit (10)) | 53184 | 2 | 20 | 1 | 11 |
| Mini-DeiT-Ti Patch16-224 (original) | 3090304 | 12 | 192 | 3 | 4 |

## A.5 Additional Representational Analysis Results

**Linear CKA results for additional tasks.** This subsection extends the representational analysis in Section 5.3. Linear Centered Kernel Alignment (Linear CKA) (Kornblith et al., 2019) measures similarity between two layers' representation matrices, with values in $[0, 1]$ (larger means more similar). In the heatmaps, a late-layer plateau (consistently high similarity among upper layers) suggests weaker depth-wise differentiation, while a similarity that decays with layer distance suggests stronger depth-wise differentiation.

In the main results, some tasks show QKOV-LoRA higher than LoRA with a Holm-corrected significance mark, while others show QKOV-LoRA lower than LoRA with a Holm-corrected significance mark (Tables 2 and 3). Below, we group additional CKA heatmaps by whether QKOV-LoRA is higher than LoRA with a significance mark, lower than LoRA with a significance mark, or has no significance mark. We also list the corresponding accuracies (mean $\pm$ std) for LoRA and QKOV-LoRA.

QKOV-LoRA is higher than LoRA on QQP ($85.1 \pm 0.20$ vs $87.6 \pm 0.15^{\dagger}$), BoolQ ($70.1 \pm 1.0$ vs $72.6 \pm 0.10^{\dagger}$), and CB ($66.7 \pm 0.80$ vs $81.0 \pm 2.2^{\dagger}$), with a significance mark versus LoRA in each case. On BoolQ and CB, the significance marks against LoRA appear only for QKOV-LoRA: QK/OV-Circuit is $69.9 \pm 0.58$ on BoolQ and $71.1 \pm 3.8$ on CB (no significance marks).

QKOV-LoRA is lower than LoRA on MRPC ($90.7 \pm 0.95$ vs $86.4 \pm 1.2^{\dagger}$), MNLI ($84.1 \pm 0.15$ vs $82.7 \pm 0.16^{\dagger}$), QNLI ($91.6 \pm 0.24$ vs $90.7 \pm 0.19^{\dagger}$), and MultiRC ($77.4 \pm 0.40$ vs $74.3 \pm 0.50^{\dagger}$), with a significance mark versus LoRA in each case.

On SST-2, GLUE RTE, COPA, and SuperGLUE RTE, QKOV-LoRA does not differ significantly from LoRA (SST-2: $91.8 \pm 0.40$ vs $91.6 \pm 0.23$; RTE (GLUE): $72.7 \pm 2.3$ vs $72.9 \pm 1.2$; COPA: $61.7 \pm 4.6$ vs $63.0 \pm 1.6$; RTE (SuperGLUE): $70.6 \pm 2.5$ vs $72.0 \pm 1.5$). For WSC, QKOV-LoRA is $53.8 \pm 2.8$ compared to $58.0 \pm 3.3$ for LoRA, without a significance mark.

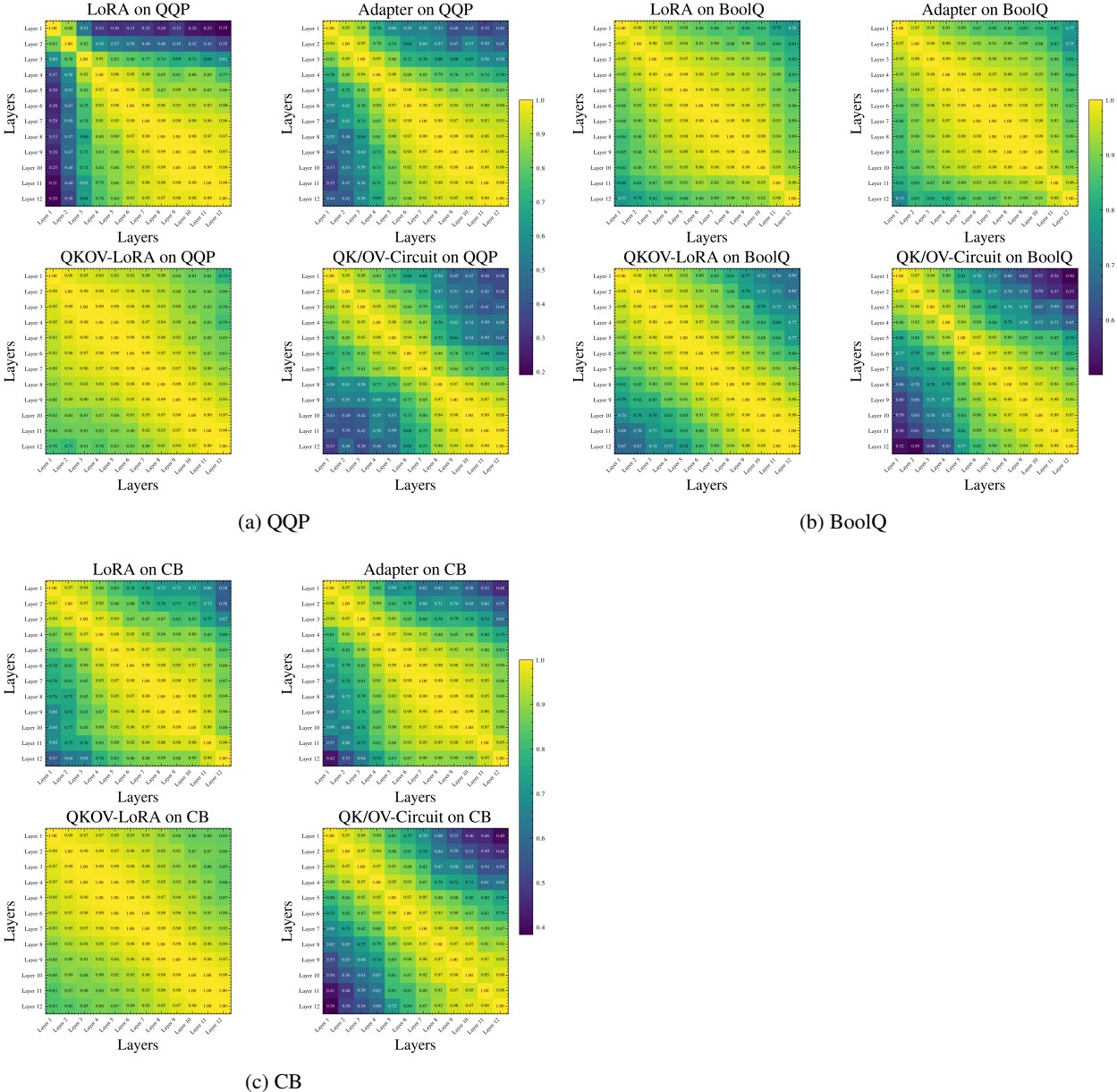

Figure 10: Linear CKA heatmaps for tasks where QKOV-LoRA is higher than LoRA with a significance mark. Panels correspond to (a) QQP, (b) BoolQ, and (c) CB.

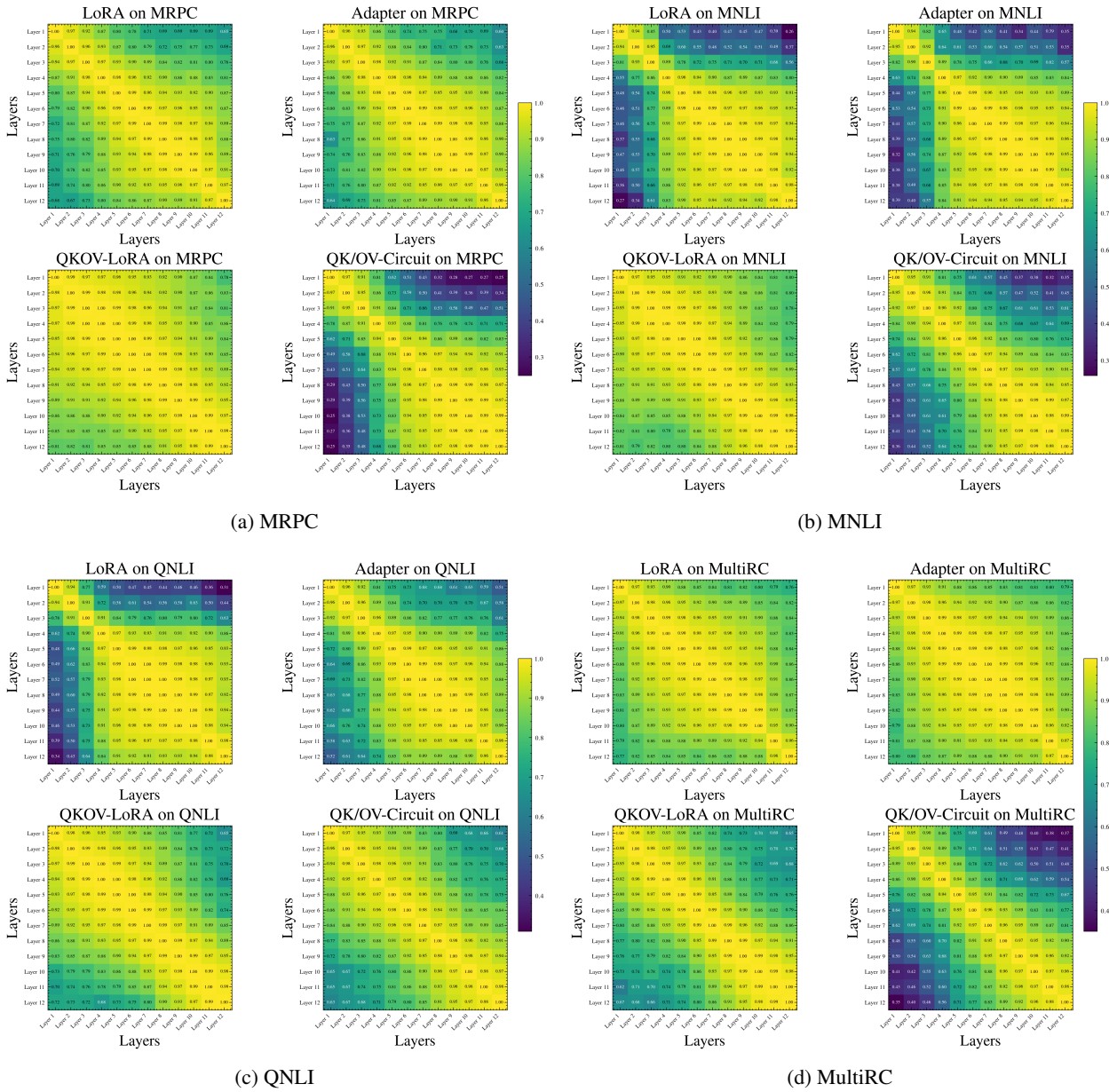

Figure 11: Linear CKA heatmaps for tasks where QKOV-LoRA is significantly lower than LoRA. Panels correspond to (a) MRPC, (b) MNLI, (c) QNLI, and (d) MultiRC.

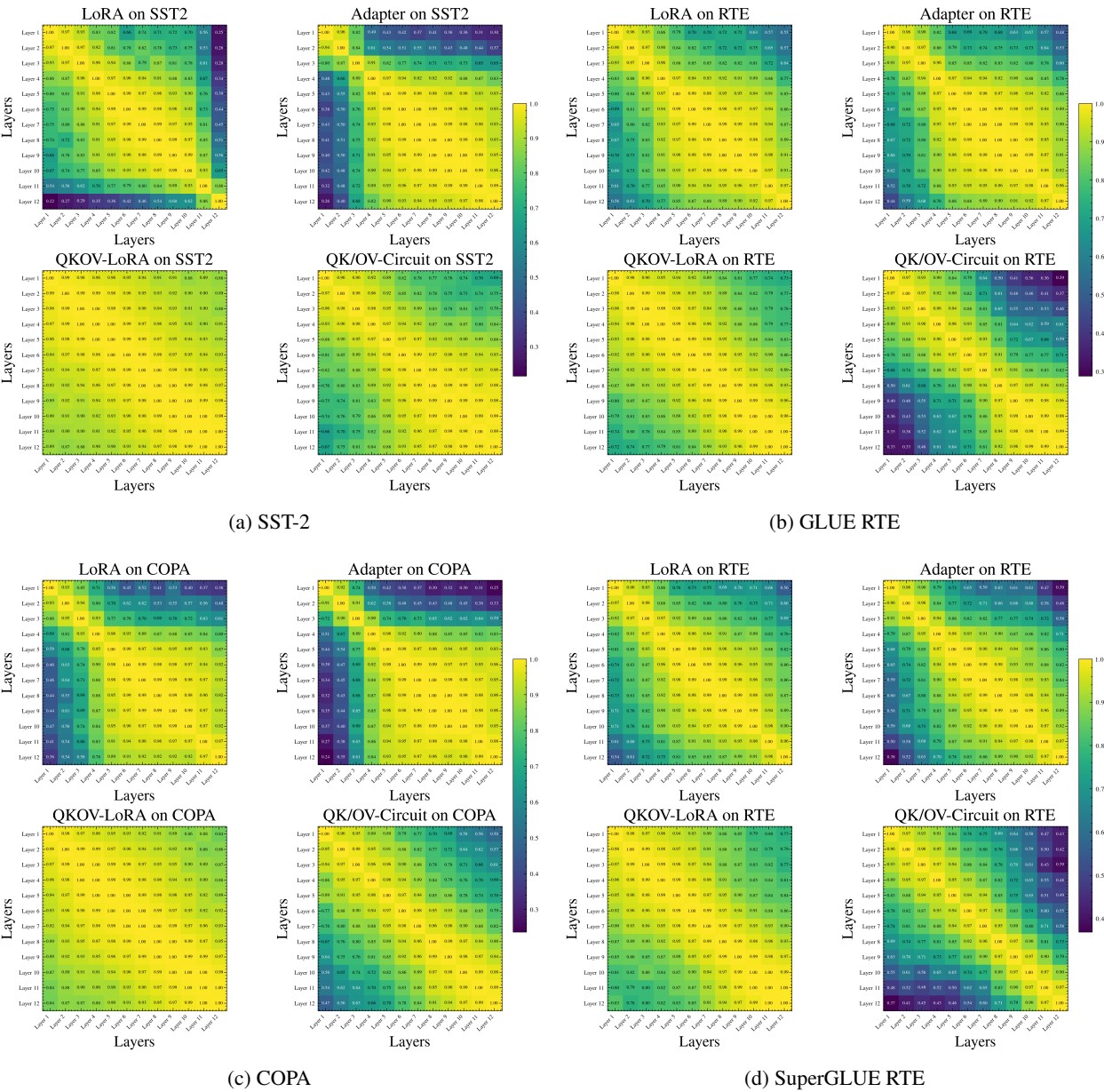

(a) SST-2

(b) GLUE RTE

(c) COPA

(d) SuperGLUE RTE

Figure 12: Additional Linear CKA heatmaps for tasks where QKOV-LoRA does not differ significantly from LoRA (SST-2, GLUE RTE, COPA, SuperGLUE RTE). Panels correspond to (a) SST-2, (b) GLUE RTE, (c) COPA, and (d) SuperGLUE RTE.

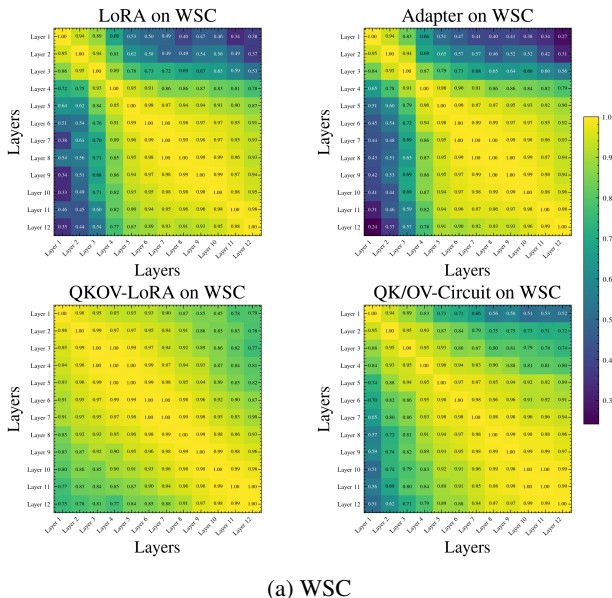

(a) WSC

Figure 13: Additional Linear CKA heatmap for WSC, where QKOV-LoRA does not differ significantly from LoRA. Panel corresponds to (a) WSC.

