# OpenReview forum: "Depth as Modulation in Weight-Sharing Transformers"
_TMLR — Rejected by TMLR_

### Review · Reviewer_ky7n · 2025-12-16

**Summary Of Contributions:**

This paper studies weight sharing in Transformer models, as used in approaches such as the Universal Transformer and ALBERT, where a single Transformer block is repeatedly applied across multiple layers. The paper argues that strict repetition is undesirable because the same transformation is enforced at every depth, which limits depth-dependent behavior.

To address this, the paper proposes adding small, depth-specific trainable perturbations to a shared Transformer block, while keeping the base weights fixed. These perturbations are applied only to the self-attention module. The paper explores how performance varies with the placement and depth of these perturbations.

Two instantiations are considered: QKOV-LoRA, which applies per-layer low-rank updates to the Q/K/V/O projections, and QK/OV-circuit, which applies low-rank corrections directly to the QK attention logits and the OV value aggregation. Experiments on ALBERT (SuperGLUE) and ViT-B/16 (ImageNet and transfer tasks) show that depth-varying perturbations can outperform strict weight sharing and depth-invariant adapters in some configurations, while degrading performance on others.

**Audience:**

Yes

**Audience Explanation:**

As mentioned above, this paper is outside my area of expertise. That said, the idea of adding perturbations to a repeatedly applied fixed block appears potentially interesting for designing parameter-efficient Transformers. While I do not have sufficient familiarity with the literature to assess the true novelty of this idea, it may still attract interest from some readers in this field.

**Claims And Evidence:**

No

**Claims Explanation:**

This paper is not in my area of expertise, so my comments are based on a general reading rather than a detailed technical review.

1. The paper is difficult to read and gives a strong impression of having been written with heavy use of LLMs. The writing is overly polished and repetitive, and many sentences sound unnatural. In particular, terms such as “autonomous system,” “depth-by-perturbation principle,” and “quasi-autonomous design” are introduced and reused without being clearly defined or grounded in precise technical meaning.

2. The paper presents the “depth-by-perturbation principle” as its main contribution. However, this does not appear to be a principle in a theoretical sense, nor a clearly specified design methodology. In practice, it merely refers to the idea of adding small trainable components to a Transformer block that is otherwise repeated across layers, which is a straightforward and previously explored approach (see point 4).

3. The experimental results are weak and inconsistent. For example, in Table 1, many reported improvements are small relative to the reported variance, and no statistical significance analysis is provided. Some tasks show degraded performance, making it difficult to draw strong conclusions.

4. The related work section appears outdated and incomplete. A brief search reveals several relevant recent papers that are not cited (e.g., https://arxiv.org/pdf/2410.20672 and https://arxiv.org/abs/2503.17750 ). At a minimum, this suggests that the idea of perturbing shared or repeated blocks is less novel than the paper claims.

**Requested Changes:**

See comments in the Evidence criteria section.

---

> ### Author Response · Authors · 2026-02-22
> **Reponse to Reviewer ky7n**
>
> We thank the reviewer for their constructive feedback. In our revision, we have simplified the writing, scaled back our claims, added statistical significance tests, and updated the related work. Detailed responses to each comment are below.
>
> ### 1: Writing clarity and terminology
>
> > The paper is difficult to read and gives a strong impression of having been written with heavy use of LLMs. The writing is overly polished and repetitive, and many sentences sound unnatural. In particular, terms such as “autonomous system,” “depth-by-perturbation principle,” and “quasi-autonomous design” are introduced and reused without being clearly defined or grounded in precise technical meaning.
>
> We completely agree that the original terminology was confusing and detracted from the paper. We have removed the phrases “autonomous system,” “depth-by-perturbation principle,” and “quasi-autonomous design” entirely. We rewrote the Abstract, Introduction, and Conclusion to describe the method strictly in concrete implementation terms: a frozen shared block with small, depth-indexed modules inside multi-head self-attention (MHSA). We also extensively edited the text to remove repetitive phrasing.
>
> **Where in the revised manuscript:** Abstract; Section 1 (Introduction); Section 6 (Conclusion).
>
> ### 2: Novelty claims and “principle” framing
>
> > The paper presents the “depth-by-perturbation principle” as its main contribution. However, this does not appear to be a principle in a theoretical sense, nor a clearly specified design methodology. In practice, it merely refers to the idea of adding small trainable components to a Transformer block that is otherwise repeated across layers, which is a straightforward and previously explored approach (see point 4).
>
> We agree that framing this as a "principle" overstated the contribution. We have removed this framing. The revision now correctly positions the work as an empirical study of a specific architecture choice for weight-sharing Transformers: keeping the shared backbone fixed and training only small, depth-indexed Low-Rank Adaptation (LoRA) modules inside the MHSA. We focus on characterizing when depth-dependent modulation helps or hurts under matched trainable-parameter budgets.
>
> **Where in the revised manuscript:** Abstract; Section 1 (Introduction); Section 6 (Conclusion).
>
> ### 3: Variability / significance / mixed results
>
> > The experimental results are weak and inconsistent. For example, in Table 1, many reported improvements are small relative to the reported variance, and no statistical significance analysis is provided. Some tasks show degraded performance, making it difficult to draw strong conclusions.
>
> We agree that the original results lacked proper statistical validation. In the revision, we now report the mean ± standard deviation over 10 random seeds for all GLUE and SuperGLUE tasks. We also added Holm-corrected significance testing compared to the LoRA baseline (†, p < 0.05).
>
> To better explain the mixed results (i.e., why some tasks improve while others degrade), we included a new representational analysis using Linear CKA (Centered Kernel Alignment) and token cosine similarity.
>
> *(Note: The main GLUE/SuperGLUE results are now in Tables 2 and 3, as we repurposed Table 1 to summarize configurations and computational costs.)*
>
> **Where in the revised manuscript:** Section 5 (Experiments); Table 2 (GLUE); Table 3 (SuperGLUE); Section 5.3 (Representational Analysis); Appendix A.5; Section 6 (Conclusion).
>
> ### 4: Missing related work
>
> > The related work section appears outdated and incomplete. A brief search reveals several relevant recent papers that are not cited (e.g., `https://arxiv.org/pdf/2410.20672` and `https://arxiv.org/abs/2503.17750` ). At a minimum, this suggests that the idea of perturbing shared or repeated blocks is less novel than the paper claims.
>
> Thank you for pointing out these missing references. We have added them to Section 2. We also expanded the discussion to clarify the differences between our approach and these recent papers—specifically, the distinctions between end-to-end retraining versus freezing the shared block, and within-layer sharing of low-rank modules versus across-depth modulation.
>
> **Where in the revised manuscript:** Section 2 (Related Work).
>
> ---
>
> We hope these changes address your concerns. Please let us know if any further clarifications are needed.

---

### Review · Reviewer_gXdR · 2026-01-10

**Summary Of Contributions:**

This paper studies the efficiency–expressivity tension in weight-sharing Transformers: repeating a single block saves parameters but can undermine the depth-wise functional specialization needed for iterative refinement. The authors propose “depth-by-perturbation”: repeatedly apply a shared (frozen) block while injecting small, layer-indexed trainable perturbations into MHSA so the effective update rule varies with depth.

Strengths:
1) Clear motivating principle + consistent framing. The non-autonomous dynamics viewpoint provides a coherent story for why strict sharing can be limiting and why depth-varying modulation should help.
2) Cross-domain evaluation with language tasks and vision tasks.
3) Useful vision ablation motivating “perturb MHSA, share FFN”

Weaknesses see the below.

**Additional Comments:**

Overall: Weak Accept / Borderline.
I find the core idea clearly motivated, and the cross-domain experiments are promising. However, the paper needs stronger ablations to isolate the effect of depth-variation from parameter/capacity effects and clearer experimental reporting (ranks, parameter counts, and the 10% ImageNet protocol-5.3.3 & 5.3.4).

**Audience:**

Yes

**Audience Explanation:**

The paper solves a timely, practical problem—making Transformers more parameter-efficient without giving up the benefits of depth—using a simple recipe: weight sharing plus small, depth-dependent perturbations. It tests the idea in both NLP and vision and includes ablations that give useful design guidance. This should be of interest to TMLR readers working on efficient architectures, parameter sharing, and parameter-efficient finetuning.

**Broader Impact Concerns:**

Might add a statement that to be related to environmental impact.

**Claims And Evidence:**

No

**Claims Explanation:**

1) The core causal claim (“depth variation matters”) is not cleanly isolated. Without matched-parameter controls (depth-varying vs depth-invariant with equal trainable params), it’s hard to conclude improvements come specifically from depth dependence rather than simply adding more effective capacity.
2) The experimental reporting is unclear. Rank choices appear inconsistent across sections, which makes fairness hard to assess—please state the rank used in each experiment explicitly and justify it with baselines. Also, the 10% ImageNet setting (Secs. 5.3.3–5.3.4) lacks comparisons to closely related methods under the same protocol.
3) NLP results are mixed and don’t uniformly support the broad claim. SuperGLUE shows improvements vs some baselines, but also meaningful regressions on certain tasks (e.g., WiC/WSC drops in some configurations). This is not necessarily a deal-breaker, but it weakens any strong claim of consistent advantage.
4) Mechanistic/interpretive claims are mostly untested. The paper motivates the approach via “iterative refinement” / specialization ideas, but does not provide direct representational or behavioral evidence (e.g., depth-wise specialization metrics). It frames these analyses as future work.

**Requested Changes:**

The following changes are critical for acceptance:
1. Clarify and standardize reporting. Put ranks, trainable parameter counts, and recurrence start index (l) in every results table/figure caption (or a single consolidated table).
2. Add a “matched-parameter” ablation to isolate the effect of depth-variation.
3. Mechanistic evidence of specialization. Even lightweight analyses for your proposed method and baselines would strengthen the “depth as modulation” story and distinguish it from “just another adapter”
4. Broaden NLP evaluation. If feasible, include additional benchmarks with stronger comparisons to other PEFT baselines.

The following is to simply strengthen the work:
Simplifying the writing and using more straightforward wording throughout would make the paper easier to follow.

---

> ### Author Response · Authors · 2026-02-22
> **Reponse to Reviewer gXdR**
>
> We thank you for the constructive feedback and the clear requested changes. In our revision, we have addressed your concerns regarding reporting, matched-parameter controls, mechanistic analysis, and claim calibration. Below is a point-by-point response indicating where each change appears in the updated manuscript.
>
> ---
>
> ### 1. Clarify and standardize reporting
> > **Comment:** *Clarify and standardize reporting. Put ranks, trainable parameter counts, and recurrence start index (l) in every results table/figure caption (or a single consolidated table).*
>
> **Response:** We have standardized our reporting to make fair comparisons easier. Each results table and figure caption now explicitly reports (or points to a consolidated summary of): (i) trainable parameter count, (ii) LoRA rank (where applicable), and (iii) the start index *l*. For the vision tasks, we now separate fixed-rank index sweeps from matched-parameter comparisons to make the control variables explicit. We also included matched-parameter comparisons against MiniViT under the same protocol in the 10% ImageNet setting.
>
> **Changes in revision:** Table 1; Tables 2–3 (captions refer to Table 1); Figure 8; Tables 4–5; Appendix A.2; Appendix (“MiniViT Baseline Details”).
>
> ### 2. Matched-parameter controls
> > **Comment:** *Add a “matched-parameter” ablation to isolate the effect of depth-variation.*
>
> **Response:** We have reorganized our main comparisons around matched trainable-parameter budgets to isolate the effect of depth variation from simply adding capacity. Depth-varying methods are now directly compared against depth-invariant baselines under comparable adaptation budgets in both language and vision settings.
>
> **Changes in revision:** Table 1; Tables 2–4; Table 5; Appendix A.2.
>
> ### 3. Mechanistic evidence of specialization
> > **Comment:** *Mechanistic evidence of specialization. Even lightweight analyses for your proposed method and baselines would strengthen the “depth as modulation” story and distinguish it from “just another adapter”.*
>
> **Response:** To provide mechanistic evidence beyond task performance, we added a representation-level analysis using Linear Centered Kernel Alignment (CKA) and within-layer token cosine similarity. We report these metrics on two contrasting tasks (CoLA and WiC) to compare a scenario where depth-indexed modulation helps versus one where it underperforms a depth-invariant baseline. This analysis helps distinguish our approach from standard adapters.
>
> **Changes in revision:** Section 5.3; Figures 3–6; Appendix A.5.
>
> ### 4. Broaden NLP evaluation and calibrate claims
> > **Comment:** *Broaden NLP evaluation. If feasible, include additional benchmarks with stronger comparisons to other PEFT baselines.*
>
> **Response:** We have expanded our NLP evaluation to include GLUE and SuperGLUE. The results now report the mean ± standard deviation over 10 seeds, with Holm-corrected significance testing against the LoRA baseline (*p* < 0.05). We also added standard Adapters as an additional parameter-efficient baseline. Finally, we revised the Abstract and Conclusion to better reflect these task-dependent outcomes, explicitly acknowledging regressions on certain tasks.
>
> **Changes in revision:** Tables 2–3; Abstract; Section 6 (Conclusion).
>
> ### 5. Writing simplification
> > **Comment:** *Simplifying the writing and using more straightforward wording throughout would make the paper easier to follow.*
>
> **Response:** We have revised the text to be more direct and concise. We removed repetitive phrasing and now describe the method in clearer implementation terms. We also adopted more modest framing in sections where results are mixed.
>
> **Changes in revision:** Abstract; Section 1 (Introduction); Section 6 (Conclusion).
>
> ### 6. Broader Impact statement
> > **Comment:** *Broader Impact Concerns: Might add a statement that to be related to environmental impact.*
>
> **Response:** We added a Broader Impact statement discussing the environmental implications of our work. It highlights potential reductions in compute and memory requirements during fine-tuning, while also noting that compression and parameter sharing can impact performance unevenly across tasks and should be monitored in deployment.
>
> **Changes in revision:** “Broader Impact Statement” section.
>
> ---
>
> We hope these revisions fully address your concerns and improve the clarity of the paper.

---

### Review · Reviewer_pRnD · 2026-02-08

**Summary Of Contributions:**

This paper proposes "depth-by-perturbation," i.e introducing small, learnable, layer-indexed perturbations to the self-attention module of an encoder block to improve weight-sharing transformer models such as ALBERT. The authors frame the problem as a dynamical system, with standard weight-sharing models imposing an autonomous update rule, whereas the proposed perturbations would allow depth-wise functional specialization. Two mechanisms are introduced: (1) QKOV-LoRA, which applies independent LoRA modules to all four projection matrices in the attention operation; and (2) QK/OV-circuit, a more parsimonious design that perturbs the composite QK (read) and OV (write) circuits directly, adopting intuitions from mechanistic interpretability literature. The methods are evaluated on NLP (SuperGLUE using ALBERT-base-v2) and computer vision (ImageNet/CIFAR-10 using ViT-B/16), showing improvements over traditional LoRA and reduced overfitting in the vision domain.

The main strengths of the work are:

- The conceptual framing is well-motivated and clearly presented, with the dynamical systems perspective connecting autonomous vs. non-autonomous systems to weight-sharing vs. depth-varying architectures acting as an intuitive justification for the proposed approach. The separation of QK (routing) and OV (aggregation) circuits as distinct perturbation targets is principled and draws meaningfully from prior interpretability work (Elhage et al., 2021).

- The motivational analysis in Section 5.3.1 (Figure 3) convincingly shows that sharing MHSA causes far greater performance degradation than sharing FFN, validating the design decision to target MHSA for perturbation while freezing FFN.

- The experimental setup is careful in its choice of baselines for the vision experiments: comparing against both pure weight sharing and depth-invariant LoRA adapters isolates the specific contribution of depth-varying perturbations from the general effect of adding trainable parameters.

- The experimentation is carried out in a robust and reproducible way, thanks to the hyperparameter sweep and the reporting of mean ± standard deviation over multiple seeds.

- The QK/OV-circuit method shows notably better training stability (lower variance) than QKOV-LoRA, especially in the low-data regime and at aggressive compression levels, which is a practically useful finding.

The main weaknesses of this work are:

- The NLP evaluation is limited to ALBERT-base-v2, which is a small and dated model. The generalizability of the depth-by-perturbation principle to more modern and larger weight-sharing or recurrent architectures is not demonstrated. Moreover, the use of different $r$ values for baseline LoRA and the proposed methods, while supporting the claim that strategic allocation matters more than capacity, can confound comparisons of methods' performance at a fixed rank. Testing the proposed methods at r=16 would allow authors to disentangle the effects of depth-varying allocation from rank selection, ensuring that the chosen method performs well even for higher rank values.

- The vision experiments include a useful motivational analysis and low-data evaluation, but lack comparison against other parameter-efficient methods for ViTs beyond simple weight sharing and depth-invariant LoRA. Methods such as Visual Prompt Tuning (Jia et al., 2022, which is cited but not compared against) or MiniViT (Zhang et al., 2022, also cited) would provide a more complete picture of where depth-by-perturbation stands among existing compression and adaptation techniques.

- The paper reports mixed results on NLP tasks, with degradation on WiC, WSC, and MultiRC, and modest overall average improvements (+1.1 points for the best QKOV-LoRA configuration). The explanation offered, i.e., that WSC requires localized computations while CB/BoolQ benefits from iterative refinement, is speculative and not supported by additional analysis.

- The paper lacks any time complexity analysis, which would be greatly helpful in assessing the practical usability of the proposed method. While parameter counts are reported, the computational overhead of applying depth-indexed perturbations (especially for QK/OV-circuit, which modifies composite operators) is not quantified. This omission makes it difficult to fully assess the practical efficiency claims.

**Audience:**

Yes

**Audience Explanation:**

The tension between parameter sharing and depth-wise specialization is a real and underexplored problem. Weight-sharing models are currently rarely adopted in practice, and principled methods for reintroducing functional diversity within the sharing paradigm remain scarce. The depth-by-perturbation principle provides a clean conceptual framework that could inspire further research in this area, and the tunable trade-off between the recurrence start index l and model performance (clearly visible in Figures 4 and 5) is an appealing property for practitioners. From a conceptual standpoint, the connection to Transformer Circuits (Elhage et al., 2021) offers a novel way to leverage the residual-stream view of transformer models for targeted improvements. In particular, the QK/OV-circuit design, which directly targets the functional decomposition of attention into read/write operations, is a novel contribution that validates theoretical findings. Finally, the finding that QK/OV-circuit outperforms or matches QKOV-LoRA with fewer parameters, particularly under challenging conditions (low data, aggressive compression), challenges the intuition that more fine-grained control is always better and may inform future work on efficient adaptation strategies.

**Broader Impact Concerns:**

The paper does not include a Broader Impact statement. The work is primarily a methodological contribution to parameter-efficient Transformer design and does not raise particular ethical concerns. However, a brief statement may be appropriate to highlight the potential for compressed models to be deployed in resource-constrained settings.

**Claims And Evidence:**

No

**Claims Explanation:**

The paper makes three main claims. The first, that depth-by-perturbation effectively overcomes the limitations of standard parameter sharing, receives partial support. In the vision domain, the evidence is reasonably strong: the motivational analysis clearly demonstrates MHSA's sensitivity to sharing, and both proposed methods substantially outperform depth-invariant baselines. However, in NLP the evidence is weaker: the best QKOV-LoRA configuration (l=8) achieves only a +1.1 average improvement over the LoRA baseline, and this average is driven largely by the CB task (+14.3) while several other tasks degrade (MultiRC: -3.1, WiC: -4.4, WSC: -4.2). Table 3 in the Appendix further reveals that performance is highly sensitive to the choice of start index l, with no clear pattern in the optimal selection across tasks. The claim of general effectiveness is therefore not convincingly supported for NLP.

The second claim, regarding the relative merits of QKOV-LoRA and QK/OV-circuit, is adequately supported. The low-data experiments (Table 2) and CIFAR-10 transfer (Figure 5) consistently show that the QK/OV circuit's superior stability and competitive accuracy align with the authors' interpretation of a beneficial regularization effect from the more constrained design.

The third claim regarding robustness in data-scarce scenarios is supported by Table 2; however, the results are difficult to interpret in isolation. The paper reports accuracy at various compression levels but does not compare against other parameter-efficient methods in the same low-data setting. Without such comparisons, it is unclear whether depth-by-perturbation offers advantages over alternative approaches to efficient adaptation under data scarcity.

An additional concern is that the paper frames its contribution as partly reconciling parameter efficiency with depth-wise specialization, yet it provides no direct evidence that the learned perturbations induce meaningful functional specialization across depth. An analysis of the learned perturbation matrices (e.g., via singular value decomposition, attention pattern visualization, or probing tasks at different effective depths) would substantially strengthen this central narrative.

**Requested Changes:**

Main revisions:

- The comparison between the proposed methods (r=4) and the LoRA baseline (r=16) in the NLP experiments conflates two variables: depth-varying vs. depth-invariant allocation, and rank. To isolate the contribution of depth-wise variation, the authors should include results for QKOV-LoRA and QK/OV-circuit at r=16 (matching the baseline rank), or alternatively, include the LoRA baseline at r=4 (matching the proposed methods' rank).

- The paper should include comparisons against at least one other established parameter-efficient adaptation method for ViTs (e.g., Visual Prompt Tuning, or MiniViT) in the vision experiments. This is necessary to contextualize the depth-by-perturbation approach within the broader landscape of efficient Transformer design.

- An analysis of how the learned perturbations evolve across depth, e.g., through visualization of attention patterns at different effective layers, singular value analysis of the perturbation matrices, or CKA similarity between representations at successive depths (comparing the proposed methods against pure weight sharing), would be needed to substantiate the proposed dynamical systems framing.

Other changes:

- Report wall-clock training and inference times, and FLOPs, for all methods. The paper discusses parameter counts but efficiency is mainly measured in speed for practical purposes. This verifies that the proposed method does not introduce non-trivial computational overhead despite having fewer parameters.

- The explanation for why the proposed methods degrade on WiC and WSC (Section 5.2) is speculative. The authors should either provide supporting evidence or present this as an open limitation rather than an explanation. More generally, the high sensitivity of results to the start index in Table 3 warrants further discussion: is there a principled way to select the index for a given task, or does it require exhaustive tuning?

- The paper would benefit from evaluating on a more modern weight-sharing or recurrent language model beyond ALBERT-base-v2 (e.g., recent recurrent architectures like RWKV [3] or Mamba [4]) to demonstrate that the depth-by-perturbation principle generalizes beyond a single, dated architecture. If this is infeasible, the limitation should be explicitly acknowledged.

- In Section 4.3.2, the shapes provided imply that the perturbation cost scales with sequence length, which is unusual for a parameter-efficient method and warrants clarification. Is this the intended design, and how does it affect applicability to variable-length inputs?

---

> ### Author Response · Authors · 2026-02-22
> **Reponse to Reviewer pRnD**
>
> We thank the reviewer for their constructive feedback and for highlighting the strengths of our work, including the motivation for targeting MHSA and our experimental design. Based on your comments, we have revised the manuscript to include: (i) clearer controls separating depth-wise variation from capacity, (ii) a broader vision baseline, (iii) new representational analyses, and (iv) explicit reporting of computational costs.
>
> Detailed responses to your specific points are provided below.
>
> ---
>
> ## Main revisions
>
> ### Rank control in NLP (rank vs. depth variation)
>
> > The comparison between the proposed methods (r=4) and the LoRA baseline (r=16) in the NLP experiments conflates two variables: depth-varying vs. depth-invariant allocation, and rank. To isolate the contribution of depth-wise variation, the authors should include results for QKOV-LoRA and QK/OV-circuit at r=16 (matching the baseline rank), or alternatively, include the LoRA baseline at r=4 (matching the proposed methods' rank).
>
> Fixed-rank comparisons are indeed a useful diagnostic for LoRA-based methods. While our main comparisons span different adaptation mechanisms—where trainable parameter count serves as a more consistent metric for "adaptation budget"—we have added explicit LoRA rank reporting throughout the revision.
>
> To directly address your request for a rank-matched reference, we have included a **rank-4 LoRA baseline** and Start Index sweeps in the Appendix (SuperGLUE development set). For the vision experiments, we also added a fixed-rank Start Index sweep (rank 16) to isolate its effect under a matched protocol.
>
> **Updated sections:** Table 1; Tables 2–3 (captions refer to Table 1); Appendix Table 6; Figure 8.
>
> ---
>
> ### Vision baselines (comparison to other parameter-efficient ViT methods)
>
> > The paper should include comparisons against at least one other established parameter-efficient adaptation method for ViTs (e.g., Visual Prompt Tuning, or MiniViT) in the vision experiments. This is necessary to contextualize the depth-by-perturbation approach within the broader landscape of efficient Transformer design.
>
> To better contextualize our approach, we have added **MiniViT** as an efficiency-oriented baseline in the vision experiments. We compare our method against MiniViT under matched trainable-parameter budgets to cleanly isolate the contribution of depth-indexed modulation from overall adaptation capacity.
>
> **Updated sections:** Tables 4–5; Appendix (“MiniViT (Mini-DeiT-Ti) Baseline Details”).
>
> ---
>
> ### Evidence for depth-wise variation (representational analyses)
>
> > An analysis of how the learned perturbations evolve across depth, e.g., through visualization of attention patterns at different effective layers, singular value analysis of the perturbation matrices, or CKA similarity between representations at successive depths (comparing the proposed methods against pure weight sharing), would be needed to substantiate the proposed dynamical systems framing.
>
> We completely agree that representation-level evidence is critical. We have added a new section providing representational analyses for the language setting, utilizing (i) **Linear CKA (Centered Kernel Alignment)** across layers and (ii) **token cosine similarity** distributions. We focused on two contrasting tasks (CoLA and WiC) to illustrate scenarios where depth-indexed modulation succeeds versus where it underperforms. Additional Linear CKA heatmaps are available in the Appendix.
>
> **Updated sections:** Section 5.3; Figures 3–6; Appendix (“Additional Representational Analysis Results”).
>
> ---

---

> > ### Author Response · Authors · 2026-02-22
> > **Reponse to Reviewer pRnD**
> >
> > ## Other changes
> >
> > ### Wall-clock training/inference time and FLOPs
> >
> > > Report wall-clock training and inference times, and FLOPs, for all methods. The paper discusses parameter counts but efficiency is mainly measured in speed for practical purposes. This verifies that the proposed method does not introduce non-trivial computational overhead despite having fewer parameters.
> >
> > We have expanded our computational cost reporting to address this. For the language setting (Table 1), we now report **GFLOPs**, **inference throughput**, and **training time** (min/epoch), alongside trainable parameters. We explicitly discuss the higher compute cost of QK/OV-Circuit under this setup as an efficiency trade-off. For the vision tasks, we added **GFLOPs** and trainable parameters to the main result tables as proxy efficiency metrics.
> >
> > **Updated sections:** Table 1; Tables 4–5.
> >
> > ---
> >
> > ### WiC/WSC degradations and Start Index sensitivity (avoid speculation)
> >
> > > The explanation for why the proposed methods degrade on WiC and WSC (Section 5.2) is speculative. The authors should either provide supporting evidence or present this as an open limitation rather than an explanation. More generally, the high sensitivity of results to the start index in Table 3 warrants further discussion: is there a principled way to select the index for a given task, or does it require exhaustive tuning?
> >
> > We have revised Section 5.2 to remove speculative causal explanations, now presenting the degradations strictly as task-dependent, statistically significant differences. We also explicitly acknowledge Start Index selection as a limitation, noting that the optimal choice currently requires task-specific sweeps.
> >
> > Furthermore, the new representational analyses (Section 5.3) provide evidence-based diagnostics for layer-wise behavior, and we have added a Start Index sweep table for SuperGLUE in the Appendix to transparently show this sensitivity.
> >
> > **Updated sections:** Tables 2–3; Section 5.3; Appendix Table 6; Section 6 (Conclusion/Limitations).
> >
> > ---
> >
> > ### Generalization beyond ALBERT-base-v2 (RWKV/Mamba)
> >
> > > The paper would benefit from evaluating on a more modern weight-sharing or recurrent language model beyond ALBERT-base-v2 (e.g., recent recurrent architectures like RWKV [3] or Mamba [4]) to demonstrate that the depth-by-perturbation principle generalizes beyond a single, dated architecture. If this is infeasible, the limitation should be explicitly acknowledged.
> >
> > We acknowledge this limitation. While we did not evaluate on additional language-model families in this revision, we have updated the Conclusion to explicitly state this constraint. We also clarify that our modulation interface is specifically designed for MHSA-based Transformers and would require significant redesign to apply to non-attention recurrent architectures like RWKV and Mamba.
> >
> > **Updated sections:** Section 6 (Conclusion/Limitations).
> >
> > ---
> >
> > ### Clarification on sequence-length dependence in QK/OV-Circuit (Section 4.3.2)
> >
> > > In Section 4.3.2, the shapes provided imply that the perturbation cost scales with sequence length, which is unusual for a parameter-efficient method and warrants clarification. Is this the intended design, and how does it affect applicability to variable-length inputs?
> >
> > We have updated Section 4.3.2 to make this sequence-length dependence explicit. We clarified the tensor shapes at the attention-logit and OV-contraction insertion sites, noting that in QK/OV-Circuit, the learned correction factors depend on the token count. Consequently, these parameters are tied to the training sequence length, which complicates transferability to different lengths.
> >
> > **Updated sections:** Section 4.3.2; Section 6 (Conclusion/Limitations).
> >
> > ---
> >
> > ### Broader Impact statement
> >
> > > Broader Impact Concerns:
> > > The paper does not include a Broader Impact statement. The work is primarily a methodological contribution to parameter-efficient Transformer design and does not raise particular ethical concerns. However, a brief statement may be appropriate to highlight the potential for compressed models to be deployed in resource-constrained settings.
> >
> > We have added a Broader Impact statement to the revised manuscript. It highlights the potential of our method to reduce adaptation costs in resource-constrained settings, while also cautioning that compression and sharing mechanisms can unevenly affect task performance, necessitating careful evaluation before deployment.
> >
> > **Updated sections:** “Broader Impact Statement” section.
> >
> > ---
> >
> > We hope these revisions and clarifications address your concerns. We are happy to answer any further questions during the discussion period.

---

### Decision · Action_Editor_bEr3 · 2026-06-07

**Recommendation:** Reject

**Audience:**

Yes

**Audience Explanation:**

Reviewers agree that this paper addresses a timely problem with relevance to practioners. It falls well within the subject areas of interest for TMLR.

**Claims And Evidence:**

No

**Claims Explanation:**

After considering all final reviewer evaluations, I unfortunately cannot recommend this paper for acceptance to TMLR. Following one reviewer’s suggestion, I would, however, invite a re-submission following a major revision.

The paper presents a new method for adaptation in transformers. This method is evaluated on a range of benchmarks in vision and language, but without showing a clear advantage of the method across a range of tasks, and with limited empirical depth to quantify as an empirical contribution that provides insights beyond a new module. For TMLR, further improvements on one of these fronts would be required.

Following the review, the authors made efforts to make the claims more precise and extend their experimental results. However, the shared reviewer concerns remain, e.g. regarding evaluation protocol (e.g. parameter matching experiments) and in particular about the NLP results in section 5.2 -- which the authors also in the abstract note as mixed.

**Resubmission Of Major Revision:**

The authors may consider submitting a major revision at a later time.